# A Huber Loss Minimization Approach to Mean Estimation under User-level Differential Privacy

**Puning Zhao**
Zhejiang Lab
pnzhao@zhejianglab.com

**Lifeng Lai**
University of California, Davis
lflai@ucdavis.edu

**Li Shen**
Sun Yat-Sen University
mathshenli@gmail.com

**Qingming Li**
Zhejiang University
liqm@zju.edu.cn

**Jiafei Wu**
Zhejiang Lab
wujiafei@zhejianglab.com

**Zhe Liu**[*]
Zhejiang Lab
zhe.liu@zhejianglab.com

## Abstract

Privacy protection of users' entire contribution of samples is important in distributed systems. The most effective approach is the two-stage scheme, which finds a small interval first and then gets a refined estimate by clipping samples into the interval. However, the clipping operation induces bias, which is serious if the sample distribution is heavy-tailed. Besides, users with large local sample sizes can make the sensitivity much larger, thus the method is not suitable for imbalanced users. Motivated by these challenges, we propose a Huber loss minimization approach to mean estimation under user-level differential privacy. The connecting points of Huber loss can be adaptively adjusted to deal with imbalanced users. Moreover, it avoids the clipping operation, thus significantly reducing the bias compared with the two-stage approach. We provide a theoretical analysis of our approach, which gives the noise strength needed for privacy protection, as well as the bound of mean squared error. The result shows that the new method is much less sensitive to the imbalance of user-wise sample sizes and the tail of sample distributions. Finally, we perform numerical experiments to validate our theoretical analysis.

## 1 Introduction

Privacy is one of the major concerns in modern data analysis. Correspondingly, differential privacy (DP) [1] has emerged as a standard framework of privacy protection. Various statistical problems have been analyzed with additional DP requirements [2–5]. Among all these problems, mean estimation is a fundamental one [6–9], which is not only useful in its own right [10], but also serves as a building block of many other tasks relying on estimating gradients, such as private stochastic optimization [11–16] and machine learning [17–23]. Existing research on DP mean estimation focuses primarily on item-level cases, i.e. each user contributes only one sample. However, in many practical scenarios, especially in recommendation systems [24–26] and federated learning [27–32], a user has multiple samples. We hope to regard them as a whole for privacy protection.

In recent years, a flurry of works focus on user-level DP [33–36]. The most popular one is the Winsorized Mean Estimator (WME) proposed in [34], which takes a two-stage approach. In the first stage, WME identifies an interval, which is small but contains the ground truth $\mu$ with high probability. In the second stage, WME clips user-wise averages to control the sensitivity and then calculates the final average with appropriate noise. This method can be extended to high dimensionality by Hadamard transform [37]. The convergence rate has been established in [34] under some ideal

---

[*]Corresponding author.

38th Conference on Neural Information Processing Systems (NeurIPS 2024).

assumptions. Despite the merit of the two-stage approach from the theoretical perspective, this method may face challenges in many realistic settings. Firstly, [34] assumes that users are balanced, which means that users have the same number of items. Nevertheless, in federated learning applications, it is common for clients (each client is regarded as a user here) to possess different numbers of samples [38–40]. Secondly, this method is not suitable for heavy-tailed distributions, which is also common in reality [41–46]. For heavy-tailed distributions, the interval generated in the first stage needs to be large enough to prevent clipping bias, which results in large sensitivity. As a result, stronger additive noise is needed for privacy protection, which significantly increases the estimation error. These drawbacks hinder the practical application of user-level DP. We aim to propose new solutions to address these challenges.

Towards this goal, in this paper, we propose a new method, which estimates the mean using Huber loss minimizer [47], and then adds noise for privacy protection. A challenge is that to determine an appropriate noise strength, it is necessary to conduct a thorough analysis of the local sensitivity that considers all possible datasets. To overcome this challenge, we divide datasets into three types, including those with no outliers, a few outliers, and many outliers, and analyze these cases separately. Based on the sensitivity analysis, we then use the smooth sensitivity framework [48] to determine the noise strength carefully.

Our method has the following advantages. Firstly, our method adapts well to imbalanced datasets, since the threshold $T_i$ of Huber loss are selected adaptively according to the sample size per user, which leads to a better tradeoff between sensitivity and bias. Secondly, our method performs better for heavy-tailed distributions, since we control sensitivity by penalizing large distances using Huber loss, which yields a smaller bias than the clipping operation. Apart from solving these practical issues, it worths mentioning that our method solves robustness (to model poisoning attacks) and privacy issues simultaneously. In modern data analysis, it is common for a system to suffer from both poisoning and inference attacks at the same time [49–51]. Consequently, many recent works focus on unified methods for item-level DP and robustness to cope with both attacks simultaneously [52–55]. To the best of our knowledge, our method is the first attempt to unify robustness and DP at user-level.

The main contributions are summarized as follows.

- We propose the Huber loss minimization approach, which finds the point with minimum Huber distance to all samples. Our method is convenient to implement and only requires linear time complexity.

- For the simplest case with balanced users, we provide a theoretical analysis, which shows that our method makes a slight improvement for bounded distributions and a significant improvement for heavy-tailed distributions over the two-stage approach.

- For imbalanced users, we design an adaptive strategy to select weights and connecting points in Huber loss, which makes our method much less sensitive to the imbalance of local sample sizes of users.

- We conduct experiments using both synthesized and real data, which also verify the effectiveness of the proposed method for imbalanced users and heavy-tailed distributions.

## 2   Related Work

**User-level DP.** [27] applies a brute-force clipping method [18] for user-level DP in federated learning. [33] made the first step towards optimal rates under user-level DP, which analyzed discrete distribution estimation problems. The popular method WME was proposed in [34], which uses the idea of two-stage approaches [55–57]. The two-stage method for user-level DP has also been extended to stochastic optimization problems [58, 59]. [60] analyzes mean estimation for boolean signals under user-level DP in heterogeneous settings. There are also some works focusing on black-box conversion from item-level DP to user-level counterparts. [35] analyzed general statistical problems, which shows that a class of algorithms for item-level DP problems having the pseudo-globally stable property can be converted into user-level DP algorithms. Following [35], [61] expanded such transformation for any item-level algorithms. [36] extends the works to smaller $m$. It is discussed in [59] that these black-box methods have suboptimal dependence on $\epsilon$. [62–64] studies user-level DP under local model.

**From robustness to DP.** Robustness and DP have close relationships since they both require the outputs to be insensitive to minor changes in input samples. There are three types of methods for conversion from robust statistics to DP. The first one is propose-test-release (PTR), which was first proposed in [65], and was extended into high dimensional cases in [66]. The second choice is smooth sensitivity [48], which calculates the noise based on the "smoothed" local sensitivity. For example, [67] designed a method to protect trimmed mean with smooth sensitivity. The third solution is inverse sensitivity [6, 68, 69], which can achieve pure differential privacy (i.e. $\delta = 0$). All these methods require a detailed analysis of the sensitivity. For some recently proposed high dimensional estimators [70–72], the sensitivity is usually large and hard to analyze. As a common method for robust statistics [47], Huber loss minimization has been widely applied in robust regression [73, 74], denoising [75] and robust federated learning [76]. Huber loss has also been used in DP [77, 78] for linear regression problems.

**Concurrent work.** After the initial submission of this paper, we notice an independent work [79], which also studies mean estimation under user-level DP (which is called person-level DP in [79]). [79] considers *directional* bound, which requires that the moment is bounded in every direction. However, we consider *non-directional* bound, which bounds the $\ell_2$ norm of a random vector. We refer to Section 1.3.1 in [79] for further discussion.

Our work is the first attempt to use the Huber loss minimization method in user-level DP. With an adaptive selection of weights and connecting points between quadratic and linear parts, our method achieves a significantly better performance for imbalanced users and heavy-tailed distributions.

## 3 Preliminaries

In this section, we introduce definitions and notations. To begin with, we recall some concepts of DP and introduce the notion of user-level DP. Denote $\Omega$ as the space of all datasets, and $\Theta$ as the space of possible outputs of an algorithm.

**Definition 1.** *(Differential Privacy (DP) [1]) Let $\epsilon, \delta \geq 0$. A function $\mathcal{A} : \Omega \to \Theta$ is $(\epsilon, \delta)$-DP if for any measurable subset $O \subseteq \Theta$ and any two adjacent datasets $\mathcal{D}$ and $\mathcal{D}'$,*

$$P(\mathcal{A}(\mathcal{D}) \in O) \leq e^{\epsilon} P(\mathcal{A}(\mathcal{D}') \in O) + \delta, \tag{1}$$

*in which $\mathcal{D}$ and $\mathcal{D}'$ are adjacent if they differ only on a single sample. Moreover, $\mathcal{A}$ is $\epsilon$-DP if (1) holds with $\delta = 0$.*

Definition 1 is about item-level DP. In this work, we discuss the case where the dataset contains multiple users, i.e. $\mathcal{D} = \{D_1, \ldots, D_n\}$, with the $i$-th user having $m_i$ samples. Considering that the sample sizes of users are usually much less sensitive [80], throughout this work, we assume that the local sample sizes $m_i$ are public information. Under this setting, user-level DP is defined as follows.

**Definition 2.** *(User-level DP [34]) Two datasets $\mathcal{D}, \mathcal{D}'$ are user-level adjacent if they differ in items belonging to only one user. In particular, if $\mathcal{D} = \{D_1, \ldots, D_n\}$, $\mathcal{D}' = \{D_1', \ldots, D_n'\}$, in which $|D_i| = |D_i'| = m_i$ for all $i$, and there is only one $i \in [n]$ such that $D_i \neq D_i'$, then $\mathcal{D}$ and $\mathcal{D}'$ are user-level adjacent. A function $\mathcal{A}$ is user-level $(\epsilon, \delta)$-DP if (1) is satisfied for any two user-level adjacent datasets $\mathcal{D}$ and $\mathcal{D}'$.*

Since $m_i$, $i = 1, \ldots, N$ are public information, in Definition 2, it is required that $|D_i| = |D_i'|$, which means that two adjacent datasets need to have the same sample sizes for all users. We then state some concepts related to sensitivity, which describes the maximum change of the output after replacing a user with another one:

**Definition 3.** *(Sensitivity) Define the local sensitivity of function $f$ as*

$$LS_f(\mathcal{D}) = \sup_{d_H(\mathcal{D}, \mathcal{D}') = 1} \|f(\mathcal{D}) - f(\mathcal{D}')\|, \tag{2}$$

*in which $d_H(\mathcal{D}, \mathcal{D}') = \sum_{i=1}^{n} \mathbf{1}(D_i \neq D_i')$ denotes the Hamming distance. The global sensitivity of $f$ is $GS_f = \sup_{\mathcal{D}} LS_f(\mathcal{D})$.*

Adding noise proportional to the global sensitivity can be inefficient, especially for user-level problems. In this work, we use the smooth sensitivity framework [48].

**Definition 4.** *(Smooth sensitivity) $S_f$ is a $\beta$-smooth sensitivity of $f$, if (1) for any $\mathcal{D}$, $S_f(\mathcal{D}) \geq LS_f(\mathcal{D})$; (2) for any neighboring $\mathcal{D}$ and $\mathcal{D}'$, $S_f(\mathcal{D}) \leq e^\beta S_f(\mathcal{D}')$.*

The smooth sensitivity can be used to determine the scale of noise. In this work, the noise follows Gaussian distribution. It has been shown in [48] that if $\mathbf{W} \sim \mathcal{N}(0, (S^2(\mathcal{D})/\alpha^2)\mathbf{I})$, then the final output $f(\mathcal{D}) + \mathbf{W}$ is $(\epsilon, \delta)$-DP for the following $(\alpha, \beta)$ pair:

$$\alpha = \left\{ \begin{array}{ll} \frac{\epsilon}{\sqrt{\ln \frac{1}{\delta}}} & \text{if} \quad d = 1 \\ \frac{\epsilon}{5\sqrt{2\ln \frac{2}{\delta}}} & \text{if} \quad d > 1, \end{array} \right. \quad \beta = \left\{ \begin{array}{ll} \frac{\epsilon}{2\ln \frac{1}{\delta}} & \text{if} \quad d = 1 \\ \frac{\epsilon}{4(d+\ln \frac{2}{\delta})} & \text{if} \quad d > 1. \end{array} \right. \tag{3}$$

**Notations.** Throughout this paper, $\|\cdot\|$ denotes the $\ell_2$ norm by default. $a \lesssim b$ means that $a \leq Cb$ for some absolute constant $C$, and $\gtrsim$ is defined conversely. $a \sim b$ if $a \lesssim b$ and $b \lesssim a$.

## 4 The Proposed Method

This section introduces the algorithm structures. Details about parameter selection are discussed together with the theoretical analysis in Section 5 and 6, respectively. For a dataset $\mathcal{D} = \{D_1, \ldots, D_n\}$, in which $D_i \subset \mathbb{R}^d$ is the local dataset of the $i$-th user, denote $m_i = |D_i|$ as the sample size of the $i$-th user. Denote $N$ as the total number of samples, then $N = \sum_{i=1}^n m_i$. We calculate the user-wise mean first, i.e. $\mathbf{y}_i(\mathcal{D}) = (1/m_i) \sum_{\mathbf{x} \in D_i} \mathbf{x}$. The new proposed estimator (before adding noise) is

$$\hat{\mu}_0(\mathcal{D}) = \arg\min_{\mathbf{s}} \sum_{i=1}^n w_i \phi_i(\mathbf{s}, \mathbf{y}_i(\mathcal{D})), \tag{4}$$

in which $w_i$ is the normalized weight of user $i$, i.e. $\sum_{i=1}^n w_i = 1$. If users are balanced, then $w_i$ are the same for all $i$. $\phi_i$ is the Huber loss function:

$$\phi_i(\mathbf{s}, \mathbf{y}) = \left\{ \begin{array}{ll} \frac{1}{2}\|\mathbf{s} - \mathbf{y}\|^2 & \text{if} \quad \|\mathbf{s} - \mathbf{y}\| \leq T_i \\ T_i\|\mathbf{s} - \mathbf{y}\| - \frac{1}{2}T_i^2 & \text{if} \quad \|\mathbf{s} - \mathbf{y}\| > T_i. \end{array} \right. \tag{5}$$

$T_i$ is the connecting point between quadratic and linear parts of Huber loss. For balanced users, $w_i$ and $T_i$ are the same for all users. For imbalanced users, $w_i$ and $T_i$ are set differently depending on the per-user sample sizes. The general guideline is that $w_i$ increases with $m_i$, while $T_i$ decreases with $m_i$. The final output needs to satisfy user-level $(\epsilon, \delta)$-DP requirement. Hence, we set

$$\hat{\mu}(\mathcal{D}) = \text{Clip}(\hat{\mu}_0(\mathcal{D}), R_c) + \mathbf{W}, \tag{6}$$

in which $\text{Clip}(\mathbf{v}, R_c) = \mathbf{v} \min(1, R_c/\|\mathbf{v}\|)$ is the function that clips the result into $B_d(\mathbf{0}, R_c)$. The clipping operation is used to control the worst case sensitivity. $\mathbf{W}$ denotes the noise added to the estimated value. In this work, we use Gaussian noise $\mathbf{W} \sim \mathcal{N}(0, \sigma^2 \mathbf{I})$. The clipping radius $R_c$ is determined by the knowledge of the range of $\mu$. Given a prior knowledge $\|\mu\| \leq R$, then we can set $R_c = R$. Actually, similar to [55], our analysis shows that $R_c$ can grow exponentially with $n$ without significantly compromising the accuracy. The noise parameter $\sigma^2$ needs to be determined carefully through a detailed sensitivity analysis.

Now we comment on the implementation. As has been discussed in [76], minimizing multi-dimensional Huber loss can be implemented by a modification of an iterative Weiszfeld's algorithm [81, 82]. The overall worst-case time complexity is $O(nd/\xi)$, in which $\xi$ is the desired precision. Moreover, for bounded support, with high probability, the algorithm requires only one iteration with time complexity $O(nd)$. Details can be found in Appendix A.

## 5 Analysis: Balanced Users

In this section, to gain some insights, we focus on the relatively simpler case and assume that all users have the same number of items, i.e. $m_i$ are equal for all $i$. Therefore, throughout this section, we omit the subscript and use $m$ to denote the local sample size. Now $w_i = 1/n$ for all $i$. $T_i$ are also the same for all users, denoted as $T$ in this section. Let $\mathbf{X}$ be a random vector, whose statistical mean $\mu := \mathbb{E}[\mathbf{X}]$ is unknown. Given a dataset $\mathcal{D} = \{D_1, \ldots, D_n\}$, the goal is to estimate $\mu$, while

satisfying user-level $(\epsilon, \delta)$-DP. We present the sensitivity analysis for the general dataset $\mathcal{D}$ first, and then analyze the estimation error for a randomly generated dataset. To begin with, define

$$Z(\mathcal{D}) = \max_{i \in [n]} \|\mathbf{y}_i(\mathcal{D}) - \bar{\mathbf{y}}(\mathcal{D})\|, \tag{7}$$

in which $\bar{\mathbf{y}}(\mathcal{D}) = (1/n) \sum_{i=1}^{n} \mathbf{y}_i(\mathcal{D})$ is the overall average. A small $Z(\mathcal{D})$ indicates that the user-wise means $\mathbf{y}_i$ are concentrated within a small region. If $Z(\mathcal{D})$ is large, then there are some outliers. The sensitivity is bounded separately depending on whether the user-wise means are well concentrated. Throughout this section, we use $LS(\mathcal{D})$ to denote the local sensitivity of $\text{Clip}(\hat{\mu}_0(\mathcal{D}), R_c)$, in which the subscript $f$ in (2) is omitted.

*1) No outliers.* We first bound the local sensitivity for the case with $Z(\mathcal{D}) < (1 - 2/n)T$, in which $T$ represents $T_i$ in (5) for all $i$. It requires that all $\mathbf{y}_i$'s are not far away from their average.

**Lemma 1.** *If $Z(\mathcal{D}) < (1 - 2/n)T$, then*

$$LS(\mathcal{D}) \leq \frac{T + Z(\mathcal{D})}{n - 1}. \tag{8}$$

The proof of Lemma 1 is shown in Appendix B. Here we provide some intuition. From the definition (2), local sensitivity is the maximum change of $\hat{\mu}_0(\mathcal{D})$ after replacing some $\mathbf{y}_i$ with $\mathbf{y}_i'$. To achieve such maximum change, the optimal choice is to move $\mathbf{y}_i$ sufficiently far away in the direction of $\hat{\mu}_0(\mathcal{D}) - \mathbf{y}_i$. The impact of $\mathbf{y}_i$ on $\hat{\mu}_0(\mathcal{D})$ is roughly $Z(\mathcal{D})/(n-1)$, while the impact of $\mathbf{y}_i'$ is roughly $T/(n-1)$. Since $\mathbf{y}_i$ and $\mathbf{y}_i'$ are at opposite direction with respect to $\hat{\mu}_0(\mathcal{D})$, the overall effect caused by replacing $\mathbf{y}_i$ with $\mathbf{y}_i'$ is upper bounded by $(T + Z(\mathcal{D}))/(n-1)$.

*2) A few outliers.* Now we consider a more complex case: $Z(\mathcal{D})$ is large, and the dataset is not well concentrated, but the number of outliers is not too large. Formally, assume that there exists another dataset $\mathcal{D}^*$ whose Hamming distance to $\mathcal{D}$ is bounded by $k$, and $\mathcal{D}^*$ is well concentrated. Then we have the following lemma to bound the local sensitivity.

**Lemma 2.** *For a dataset $\mathcal{D}$, if there exists a dataset $\mathcal{D}^*$ such that $d_H(\mathcal{D}, \mathcal{D}^*) \leq k$, in which $d_H$ is the Hamming distance (see Definition 3), and $Z(\mathcal{D}^*) < (1 - 2(k+1)/n)T$, then $LS(\mathcal{D}) \leq 2T/(n-k)$.*

The proof of Lemma 2 is shown in Appendix C. The intuition is that since there exists a well concentrated dataset $\mathcal{D}^*$ with $d_H(\mathcal{D}, \mathcal{D}^*) \leq k$, $\mathcal{D}$ contains no more than $k$ outliers. At least $n - k$ other user-wise mean values fall in a small region. To achieve the maximum change of $\hat{\mu}_0(\mathcal{D})$, the optimal choice is to replace an outlier $\mathbf{y}_i$ with $\mathbf{y}_i'$, such that $\mathbf{y}_i - \hat{\mu}_0(\mathcal{D})$ and $\mathbf{y}_i' - \hat{\mu}_0(\mathcal{D})$ have opposite directions. Each of them has an effect of roughly $T/(n-k)$ on $\hat{\mu}_0(\mathcal{D})$, thus the overall change is $2T/(n-k)$.

*3) Other cases.* For all other cases, since $\|\text{Clip}(\hat{\mu}_0, R_c)\| \leq R_c$ always hold, the local sensitivity can be bounded by $LS(\mathcal{D}) \leq 2R_c$.

From the analysis above, we now construct a valid smooth sensitivity. Define

$$\Delta(\mathcal{D}) = \min \left\{ k | \exists \mathcal{D}^*, d_H(\mathcal{D}, \mathcal{D}^*) \leq k, Z(\mathcal{D}^*) < \frac{1}{2}T \right\}. \tag{9}$$

$\Delta(\mathcal{D})$ can be viewed as the number of outliers. From (9), if $\mathcal{D}$ is well concentrated, with $Z(\mathcal{D}) < T/2$, then $\Delta(\mathcal{D}) = 0$. Now we define $G(\mathcal{D}, k)$ as follows.

**Definition 5.** *(a) If $Z(\mathcal{D}) < (1 - 2/n)T$, $k = 0$, then $G(\mathcal{D}, 0) = (T + Z(\mathcal{D}))/(n-1)$;*

*(b) If conditions in (a) are not satisfied, and $\Delta(\mathcal{D})$ exists, if $k \leq n/4 - 1 - \Delta(\mathcal{D})$,*

$$G(\mathcal{D}, k) = \frac{2T}{n - k - \Delta(\mathcal{D})}; \tag{10}$$

*(c) If conditions in (a) and (b) are not satisfied, then $G(\mathcal{D}, k) = 2R_c$.*

Based on Definition 5, the smooth sensitivity is given by

$$S(\mathcal{D}) = \max_k e^{-\beta k} G(\mathcal{D}, k), \tag{11}$$

in which $\beta$ is determined in (3). Then we show that $S(\mathcal{D})$ is a valid smooth sensitivity, and the privacy requirement is satisfied.

**Theorem 1.** *With $\sigma = S(\mathcal{D})/\alpha$, in which $\alpha$ is determined in (3), $\mathbf{W} \sim \mathcal{N}(0, \sigma^2\mathbf{I})$, the estimator $\hat{\mu}$ defined in (6) is $(\epsilon, \delta)$-DP.*

To prove Theorem 1, we need to show that $S$ is a valid smooth sensitivity, i.e. two conditions in Definition 4 are satisfied. The detailed proof is provided in Appendix D.

In the analysis above, all results are derived for a general dataset $\mathcal{D}$. In the remainder of this section, we analyze the performance of estimator (6) for randomly generated samples.

## 5.1 Bounded Support

Let $\mathbf{X}$ be a random vector generated from distribution $P$ with an unknown statistical mean $\mu := \mathbb{E}[\mathbf{X}]$. We make the following assumption:

**Assumption 1.** $\mathbf{X}$ *is supported on $B_d(0, R) = \{\mathbf{u}| \|\mathbf{u}\| \leq R\} \subset \mathbb{R}^d$.*

The mean squared error is analyzed in the following theorem.

**Theorem 2.** *Let $R_c = R$, and $T = C_T R \ln(mn^3(d+1))/\sqrt{m}$ with $C_T > 16\sqrt{2/3}$. If $n > (4/\beta)\ln(nR_c/T)$, then*

$$\mathbb{E}\left[\|\hat{\mu}(D) - \mu\|^2\right] \lesssim \frac{R^2}{mn} + \frac{dR^2}{mn^2\epsilon^2}\ln(mnd)\ln\frac{1}{\delta}. \tag{12}$$

The proof of Theorem 2 is shown in Appendix E. With the selection rule of $T$ in Theorem 2, it is shown that with high probability, $\Delta(\mathcal{D}) = 0$, indicating that $\mathcal{D}$ is expected to be well concentrated around the population mean $\mu$. The smooth sensitivity $S(\mathcal{D})$ can then be bounded. The first term in the right hand side of (12) is the non-private estimation error, i.e. the error of $\hat{\mu}_0(\mathcal{D})$, while the second term is the error caused by noise $\mathbf{W}$. The condition $n > (4/\beta)\ln(nR_c/T)$ is necessary, since it ensures that $G(\mathcal{D}, k) = 2R_c$ (Definition 5 (c)) occurs only for sufficiently large $k$, thus $e^{-\beta k}$ is small, and does not affect the calculation of $S(\mathcal{D})$ in (11). A lower bound on the number of users $n$ has also been imposed for the two-stage method [34].

For the simplest case with bounded support and balanced users, the two-stage approach in [34] is already nearly optimal (Corollary 1 in [34]). Therefore, improvement in polynomial factors is impossible. Nevertheless, we still improve on the logarithm factor. The main purpose of Theorem 2 is to show that our improvement on heavy-tailed distributions and imbalanced users is not at the cost of hurting the performance under the simplest case with bounded distributions and balanced users.

## 5.2 Unbounded Support

Now we analyze the heavy-tailed case. Instead of requiring $\mathbf{X} \in B_d(0, R)$, we now assume that $\mathbf{X}$ has $p$-th bounded moment.

**Assumption 2.** *Suppose that $\mu \in B(\mathbf{0}, R)$, and the p-th $(p \geq 2)$ moment of $\mathbf{X} - \mu$ is bounded, i.e. $\mathbb{E}[\|\mathbf{X}\|^p] \leq M_p$.*

In Assumption 2, higher $p$ indicates a lighter tail and vice versa. We then show the convergence rate of mean squared error in Theorem 3.

**Theorem 3.** *Let $R_c = R$, and*

$$T = C_T \max\left\{\sqrt{\frac{1}{m}\ln\frac{3(d+1)}{\nu}}, 2(3m)^{\frac{1}{p}-1}\nu^{-\frac{1}{p}}\ln\frac{3(d+1)}{\nu}\right\}, \tag{13}$$

*with $\nu = \sqrt{d}/(n\epsilon)$ and $C_T > 8M_p^{\frac{1}{p}}$. If $n > 8(1 + (1/\beta)\ln(n/2T))$, then under Assumption 2,*

$$\mathbb{E}\left[\|\hat{\mu}(D) - \mu\|^2\right] \lesssim \frac{1}{mn} + \left[\frac{d\ln(nd)}{mn^2\epsilon^2} + \left(\frac{d}{m^2n^2\epsilon^2}\right)^{1-\frac{1}{p}}\ln^2(nd)\right]\ln\frac{1}{\delta}. \tag{14}$$

The proof of Theorem 3 is shown in Appendix F. Here we provide an intuitive understanding. From the central limit theorem, each user-wise mean $\mathbf{y}_i(\mathcal{D})$ is the average of $m$ i.i.d variables, thus it has

a Gaussian tail around the population $\mu$. However, since $\mathbf{X}$ is only required to have $p$-th bounded moment, the tail probability away from $\mu$ is still polynomial. The formal statement of the tail bound is shown in Lemma 13 in the appendix. Then the threshold $T$ is designed based on the high probability upper bound of $Z(\mathcal{D})$ to ensure that with high probability, $\Delta(\mathcal{D})$ is small. Regarding the result, we have the following remarks.

**Remark 1.** *Here we comment on small and large $m$ limits. If $m = 1$, the right hand side of* (14) *becomes $O(1/n + (d/n^2\epsilon^2)^{1-1/p})$, which matches existing analysis on item-level DP for heavy-tailed random variables [57]. For the opposite limit, with $m^{1-2/p} \gtrsim n^{2/p}\ln(nd)$, then the convergence rate is the same as the case with bounded support, indicating that the tail of sample distribution does not affect the error more than a constant factor.*

**Remark 2.** *Now we compare* (14) *with the two-stage approach. Following the analysis in [34], it can be shown that the bound of mean squared error in [34] is $\tilde{O}((d/(n^2\epsilon^2))(1/m + m^{4/p-2}n^{6/p}))$ (we refer to Appendix G for details). Therefore, we have achieved an improved rate in* (14).

The theoretical results in this section are summarized as follows. If the support is bounded, our method has the same convergence rate as the existing method. For heavy-tailed distributions, our approach significantly reduces the error, since our method avoids the clipping process. In federated learning applications, it is common for gradients to have heavy-tailed distributions [41–44], thus our method has the potential of improving the performance of federated learning under DP requirements. Apart from heavy-tailed distributions, another common characteristic in reality is that users are usually imbalanced. We analyze it in the next section.

## 6 Analysis: Imbalanced Users

Now we analyze the general case where $m_i$, $i = 1, \ldots, n$ are different. Recall that for balanced users, we have defined $Z(\mathcal{D})$ in (7) that finds the maximum distance from $\mathbf{y}_i(\mathcal{D})$ to their average $\bar{\mathbf{y}}(\mathcal{D})$. For imbalanced users, instead of taking the maximum, we define $Z_i$ separately for each $i$:

$$Z_i(\mathcal{D}) = \|\bar{\mathbf{y}}(\mathcal{D}) - \mathbf{y}_i(\mathcal{D})\|, \tag{15}$$

in which $\bar{\mathbf{y}}(\mathcal{D}) = \sum_{i=1}^{n} w_i \mathbf{y}_i(\mathcal{D})$ is the average of samples all over the dataset.

From now on, without loss of generality, suppose that users are arranged in ascending order of $m_i$, i.e. $m_1 \leq \ldots \leq m_n$. Define

$$h(\mathcal{D}, k) = \frac{\sum_{i=n-k+1}^{n} w_i(T_i + Z_i(\mathcal{D}))}{\sum_{i=1}^{n-k} w_i}. \tag{16}$$

Similar to the case with balanced users, we analyze the sensitivity for datasets with no outliers, a few outliers, and other cases separately.

*1) No outliers.* We show the following lemma.

**Lemma 3.** *If $h(\mathcal{D}, 1) \leq \min_i(T_i - Z_i(\mathcal{D}))$, then $LS(\mathcal{D}) \leq h(\mathcal{D}, 1)$.*

The general idea of the proof is similar to Lemma 1. However, the details become more complex since now the samples are unbalanced. The detailed proof is shown in Appendix H.

*2) A few outliers.* Similar to Lemma 2, we find a neighboring dataset $\mathcal{D}^*$ that is well concentrated and then bounds the local sensitivity. The formal statement is shown in the following lemma.

**Lemma 4.** *For a dataset $\mathcal{D}$, if there exists another dataset $\mathcal{D}^*$ such that $h(\mathcal{D}^*, k + 1) < \min_i(T_i - Z_i(\mathcal{D}^*))$, then $LS(\mathcal{D}) \leq 2\max_i(w_i T_i)/\sum_{i=1}^{n-k-1} w_i$.*

The proof of Lemma 4 is shown in Appendix I, which just follows the proof of Lemma 2.

*3) Other cases.* Finally, for all cases not satisfying the conditions in Lemma 3 and 4, we can just bound the local sensitivity with $2R_c$, i.e. $LS(\mathcal{D}) \leq 2R_c$.

Similar to the case with balanced users, now we define

$$\Delta(\mathcal{D}) = \min\{k | \exists \mathcal{D}^*, d_H(\mathcal{D}, \mathcal{D}^*) = k, h(\mathcal{D}^*, k_0) < \min_i(T_i - Z_i(\mathcal{D}))\}, \tag{17}$$

in which $k_0$ is any integer, and can be viewed as a design parameter. Correspondingly, the smooth sensitivity $G(\mathcal{D}, k)$ is defined as follows.

**Definition 6.** *(a) If $h(\mathcal{D}, 1) \leq \min_i (T_i - Z_i(\mathcal{D}))$, then $G(\mathcal{D}, 0) = h(\mathcal{D}, 1)$;*

*(b) If the conditions in (a) are not satisfied, and $\Delta(\mathcal{D})$ exists, then for all $k \leq k_0 - \Delta(\mathcal{D}) - 1$,*

$$G(\mathcal{D}, k) = \frac{2 \max\limits_{i \in [n]} w_i T_i}{\sum_{i=1}^{n - \Delta(\mathcal{D}) - k - 1} w_i}; \tag{18}$$

*(c) If the conditions in both (a) and (b) are not satisfied, then $G(\mathcal{D}, k) = 2R_c$.*

We still use the same settings of $\alpha$ and $\beta$ as in the case with balanced samples. With smooth sensitivity calculated using (11), the privacy requirement is satisfied:

**Theorem 4.** *Let $\mathbf{W} \sim \mathcal{N}(0, (S(\mathcal{D})^2/\alpha^2)\mathbf{I})$, in which $S(\mathcal{D}) = \max_k e^{-\beta k} G(\mathcal{D}, k)$, then the estimator $\hat{\mu}$ is $(\epsilon, \delta)$-DP.*

Now we analyze the convergence of the algorithm. We begin with Assumption 3. Intuitively, this assumption requires that the users can not be too unbalanced. At least half samples belong to users whose sample sizes are not very large.

**Assumption 3.** *Suppose there exists a constant $\gamma \geq 1$. Let $k_c = \min\{i | m_i > \gamma N/n\}$, then $\sum_{i=k_c}^{n} m_i \leq N/2$.*

In Assumption 3, $\gamma$ can be viewed as the degree of imbalance. For a better explanation, we provide the following examples:

- If users are balanced, then $\gamma = 1$;

- If the $i$-th user has $ki$ samples (which means that the number of items belonging to each user is linear in its order), then for large $n$, $\gamma$ is approximately $\sqrt{2}$.

In general, $\gamma$ is large if users are highly imbalanced. Under Assumption 3, the convergence of mean squared error is shown in Theorem 5.

**Theorem 5.** *Let the weights of users in (4) be $w_i = m_i \wedge m_c / (\sum_{j=1}^{n} m_j \wedge m_c)$, in which $m_c = \gamma N/n$. Moreover, let*

$$T_i = C_T \sqrt{\frac{R^2 \ln(Nn^2(d+1))}{m_i \wedge m_c}}, \tag{19}$$

*in which $C_T > 16\sqrt{2/3}$. In (17), $k_0 = \lfloor n/8\gamma \rfloor$.*

*With the parameters above, if $n > 8\gamma(1 + (1/2\beta)\ln(Nn))$, then under Assumption 1 and 3,*

$$\mathbb{E}\left[\|\hat{\mu}(\mathcal{D}) - \mu\|^2\right] \lesssim \frac{R^2}{N} + \frac{dR^2\gamma}{Nn\epsilon^2} \ln^2(Nnd) \ln\frac{1}{\delta}. \tag{20}$$

The proof of Theorem 5 is shown in Appendix J. If users are balanced, then $\gamma = 1$, with $N = nm$, (20) reduces to (12). From (20) and Assumption 3 it can be observed that our method is much less sensitive to a single large $m_i$. As long as $m_i$ are not very large for most users, the convergence rate of mean squared error is not affected. There are two intuitive reasons. Firstly, $w_i$ is upper bounded by $m_c/(\sum_{j=1}^{n} m_j \wedge m_c)$, thus the worst-case sensitivity is controlled. Secondly, $T_i$ are set adaptively to achieve a good tradeoff between sensitivity and bias. With larger $m_i$, smaller $T_i$ is used, and vice versa. We refer to Appendix G for comparison with the two-stage approach.

## 7 Numerical Examples

In this section, we show numerical experiments. We compare the performance of our new Huber loss minimization approach (denoted as HLM) versus the two-stage approach proposed in [34], called Winsorized Mean Estimator (denoted as WME).

## 7.1 Balanced Users

Here all users have the same sizes of local datasets. In the following experiments, we fix $\epsilon = 1$ and $\delta = 10^{-5}$. For a fair comparison, the parameter $T$ for our method as well as $\tau$ in [34] are both tuned optimally for each case. Figure 1 shows the curve of mean squared error. Let the number of users $n$ be either $1,000$ and $10,000$. In each curve, $n$ is fixed, while the number of samples per user $m$ varies from 1 to $1,000$. The results are plotted in logarithm scale. Figure 1(a)-(c) show the results of uniform distribution in $[-1, 1]$, Gaussian distribution $\mathcal{N}(0, 1)$, and the Lomax distribution, whose pdf is $f(x) = a/(1 + x)^{a+1}$ (we use $a = 4$ in Figure 1(c)). Figure 1 (d)-(f) shows the corresponding experiments with dimensionality $d = 3$. Finally, Figure 1 (g) and (h) show the results using the IPUMS dataset [83] for total income and salary, respectively, which are typical examples of data following heavy-tailed distributions.

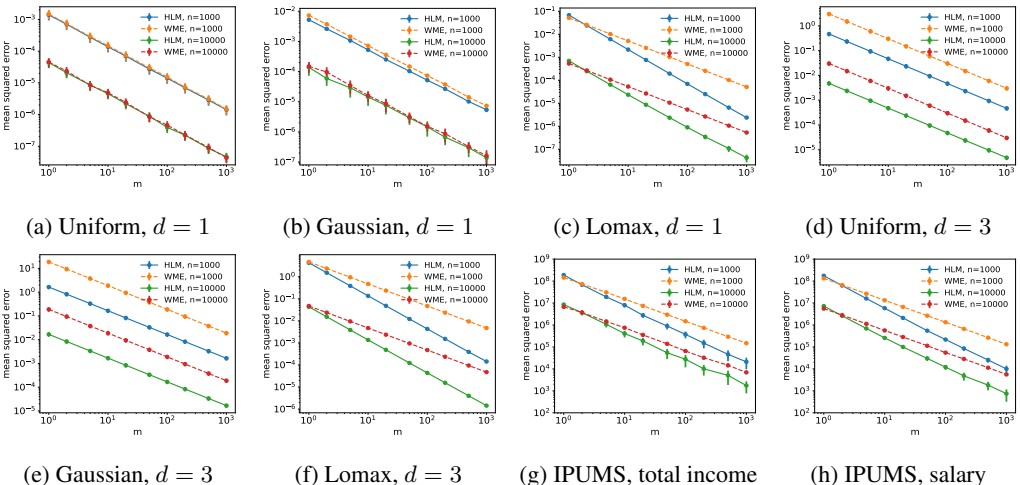

| (a) Uniform, $d = 1$ | (b) Gaussian, $d = 1$ | (c) Lomax, $d = 1$ | (d) Uniform, $d = 3$ |
|---|---|---|---|
| (e) Gaussian, $d = 3$ | (f) Lomax, $d = 3$ | (g) IPUMS, total income | (h) IPUMS, salary |

Figure 1: Convergence of mean squared error with balanced users.

Figure 1 (a) and (b) show that for one-dimensional uniform and Gaussian distribution, the Huber loss minimization approach has nearly the same performance as the two-stage method. Our explanation is that uniform and Gaussian distributions are symmetric, with no tails or light tails, thus the clipping operation does not introduce additional bias. However, for heavy-tailed and skewed distribution, such as Lomax distribution, our new method has a significantly faster convergence rate than the two-stage method. These results agree with the theoretical analysis, which shows that our method reduces the clipping bias. With higher dimensionality, Figure 1(d)-(f) show that the advantage of the practical performance of our method becomes more obvious.

## 7.2 Imbalanced Users

Now we show the performance with unbalanced users. For some $\gamma \geq 1$, let $s_i = \lceil N(i/n)^{\gamma} \rceil$ for $i = 0, \ldots, n$, and $m_i = s_i - s_{i-1}$ for $i = 1, \ldots, n$. It can be shown that Assumption 3 is satisfied with $\gamma$. Therefore, according to the analysis in Section 6, we let $w_i = m_i \wedge m_c / \sum_j m_j \wedge m_c$, with $m_c = \gamma N/n$, and $T_i = A/\sqrt{m_i \wedge m_c}$, in which $A$ is tuned optimally for each case. The selection of $T_i$ may be slightly different from (19) in Theorem 5. In Theorem 5, $T_i$ is selected to minimize the theoretical upper bound. To ensure that the analysis is mathematically rigorous, the upper bound of estimation error is larger than the truth. Therefore, the optimal value of $T_i$ in practice is slightly different from that derived in theories. Note that such parameter tuning does not require additional privacy budget since in each experiment, $T_i$ are hyperparameters that is fixed before knowing the value of each sample. They are not determined adaptively based on the data. Figure 2 shows the growth curve of mean squared error with respect to $\gamma$.

From Figure 2, it can be observed that with the increase of $\gamma$, the two-stage method degrades, while the Huber loss minimization approach performs significantly more stable.

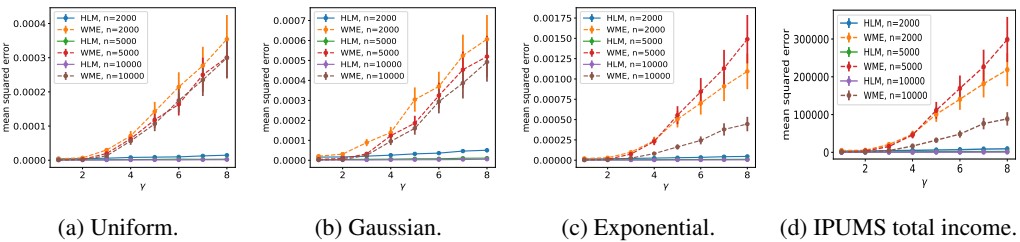

| (a) Uniform. | (b) Gaussian. | (c) Exponential. | (d) IPUMS total income. |

Figure 2: Growth of mean squared error with degree of imbalance $\gamma$.

# 8 Conclusion

In this paper, we have proposed a new approach to mean estimation under user-level DP based on Huber loss minimization. The sensitivity is bounded for all possible datasets. Based on the sensitivity analysis, we use the smooth sensitivity framework to determine the noise added to the result. We have also derived the bound on the mean squared error for various cases. The result shows that our method reduces the error for heavy-tailed distributions, and is more suitable to imbalanced users. It is promising to extend our approach to more learning problems, such as calculating average gradients in federated learning.

**Limitations:** The limitations of our work include: (1) There are some requirements on the minimum number of users $n$. while entirely removing this condition is impossible, to make our method more practical, we expect that it can be somewhat weakened. (2) The case with local sample sizes $m_i$ also being private has not been analyzed. We will leave these two points as future works.

# Acknowledgement

The work of L.Shen is supported by the STI 2030—Major Projects (No. 2021ZD0201405). The work of Z.Liu is supported by the National Natural Science Foundation of China (No.62132008 and U22B2030), Natural Science Foundation of Jiangsu Province (BK20220075). The work of L.Lai is supported by the National Science Foundation under grants ECCS-20-00415 and CCF-22-32907.

We thank Prof.Gautam Kamath for his fruitful discussions.

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

## A  Comments About Algorithm Implementation

In this section, we describe the algorithm for Huber loss minimization.

In particular, we solve

$$\mathbf{c} = \arg\min_{\mathbf{s}} \sum_{i=1}^{n} w_i \phi_i(\mathbf{s}, \mathbf{y}_i). \tag{21}$$

From (5), it can be shown that

$$\mathbf{c} = \frac{\sum_{i=1}^{n} w_i \min\left\{1, \frac{T_i}{\|\mathbf{c}-\mathbf{y}_i\|}\right\} \mathbf{y}_i}{\sum_{i=1}^{n} w_i \min\left\{1, \frac{T_i}{\|\mathbf{c}-\mathbf{y}_i\|}\right\}}. \tag{22}$$

The algorithm can be designed from above equation. Suppose that the algorithm starts from $\mathbf{c}_0$. The update rule is

$$\mathbf{c}_{k+1} = \frac{\sum_{i=1}^{n} w_i \min\left\{1, \frac{T_i}{\|\mathbf{c}_k-\mathbf{y}_i\|}\right\} \mathbf{y}_i}{\sum_{i=1}^{n} w_i \min\left\{1, \frac{T_i}{\|\mathbf{c}_k-\mathbf{y}_i\|}\right\}}. \tag{23}$$

(23) is run iteratively until the norm of update $\|\mathbf{c}_{k+1} - \mathbf{c}_k\|$ between two iterations is less than $\xi$.

We provide a brief analysis on the computational complexity as follows.

**Worst case.** [82] has shown that the Weiszfeld's algorithm for calculating geometric median needs $O(1/\xi)$ steps to achieve precision $\xi$. The proof can also be used to our algorithm (23). Moreover, from (23), each step requires $O(nd)$ time, in which $d$ is the dimensionality of $\mathbf{y}_i$, thus the overall time complexity is $O(nd/\xi)$.

**Common case.** From the analysis in Section 5 and 6, for bounded support, with high probability, $Z(\mathcal{D}) \leq T$ holds for balanced users, and $Z_i(\mathcal{D}) \leq T_i$ holds for imbalanced users. In this case, we can just calculate the result within one step:

$$\mathbf{c} = w_i \mathbf{y}_i. \tag{24}$$

Therefore the time complexity is $O(nd)$.

## B  Proof of Lemma 1

Now the users are balanced. (4) and (5) becomes

$$\hat{\mu}_0(D) = \arg\min_{\mathbf{s}} \sum_{i=1}^{n} \phi(\mathbf{s}, \mathbf{y}_i(\mathcal{D})), \tag{25}$$

and

$$\phi(\mathbf{s}, \mathbf{y}) = \begin{cases} \frac{1}{2}\|\mathbf{s}-\mathbf{y}\|^2 & \text{if} \quad \|\mathbf{s}-\mathbf{y}\| \leq T \\ T\|\mathbf{s}-\mathbf{y}\| - \frac{1}{2}T^2 & \text{if} \quad \|\mathbf{s}-\mathbf{y}\| > T. \end{cases} \tag{26}$$

In this section, we prove a strengthened version of Lemma 1 for future usage. Define the *modulus of continuity* as

**Definition 7.** *(Modulus of continuity) Define*

$$\omega(\mathcal{D}, k) = \sup_{\mathcal{D}':d_H(\mathcal{D},\mathcal{D}')\leq k} \|\hat{\mu}_0(\mathcal{D}) - \hat{\mu}_0(\mathcal{D}')\|. \tag{27}$$

From the definition, $\omega(\mathcal{D}, 0)$ is just the local sensitivity $LS(\mathcal{D})$. Then we show the following lemma.

**Lemma 5.** *If $Z(\mathcal{D}) < (1 - 2k/n)T$, then*

$$\omega(\mathcal{D}, k) \leq \frac{k(T + Z(\mathcal{D}))}{n - k}. \tag{28}$$

Let $k = 0$, then Lemma 5 reduces to Lemma 1. The remainder of this section shows the proof of Lemma 5.

According to (27), $\omega(\mathcal{D}, k)$ is the supremum change after replacing all items belonging to $k$ users. Denote $\mathcal{D}'$ as a dataset such that $d_H(\mathcal{D}, \mathcal{D}') \le k$. Let $I$ be the set of users such that $D_i \ne D_i'$. Throughout this section, we denote $\mathbf{y}_i = \mathbf{y}_i(\mathcal{D})$ and $\mathbf{y}_i' = \mathbf{y}_i(\mathcal{D}')$ for simplicity.

From (25),

$$\hat{\mu}_0(\mathcal{D}') = \arg\min_{\mathbf{s}} \left[ \sum_{i \in I} \phi(\mathbf{s}, \mathbf{y}_i') + \sum_{i \in [n] \setminus I} \phi(\mathbf{s}, \mathbf{y}_i) \right]. \tag{29}$$

Let $\nabla\phi$ be the gradient of $\phi$ with respect to the first argument. Define

$$g(\mathbf{s}) = \sum_{i \in I} \nabla\phi(\mathbf{s}, \mathbf{y}_i') + \sum_{i \in [n] \setminus I} \nabla\phi(\mathbf{s}, \mathbf{y}_i), \tag{30}$$

then

$$g(\hat{\mu}_0(\mathcal{D}')) = 0, \tag{31}$$

and

$$\begin{aligned}
g(\hat{\mu}_0(D)) &= \sum_{i \in I} \nabla\phi(\hat{\mu}_0(D), \mathbf{y}_i') + \sum_{i = [n] \setminus I} \nabla\phi(\hat{\mu}_0(D), \mathbf{y}_i) \\
&= \sum_{i \in I} \nabla\phi(\hat{\mu}_0(D), \mathbf{y}_i') - \sum_{i \in I} \nabla\phi(\hat{\mu}_0(D), \mathbf{y}_i) + \sum_{i \in [n]} \nabla\phi(\hat{\mu}_0(D), \mathbf{y}_i) \\
&= \sum_{i \in I} \nabla\phi(\hat{\mu}_0(D), \mathbf{y}_i') - \sum_{i \in I} \nabla\phi(\hat{\mu}_0(D), \mathbf{y}_i), \tag{32}
\end{aligned}$$

in which the last step comes from (25).

Then we show the following lemma.

**Lemma 6.** *If $Z(\mathcal{D}) \le T$, then $\hat{\mu}_0(\mathcal{D}) = \bar{\mathbf{y}}$, in which $\bar{\mathbf{y}} = (1/n) \sum_{i=1}^{n} \mathbf{y}_i$.*

*Proof.* The proof has two steps.

(1) $\bar{\mathbf{y}}$ **is a minimizer of** $\sum_{i=1}^{n} \phi(\mathbf{s}, \mathbf{y}_i)$**.** From (26),

$$\nabla\phi(\mathbf{s}, \mathbf{y}_i) = \begin{cases} \mathbf{s} - \mathbf{y}_i & \text{if} \quad \|\mathbf{s} - \mathbf{y}_i\| \le T \\ T\frac{\mathbf{s} - \mathbf{y}_i}{\|\mathbf{s} - \mathbf{y}_i\|} & \text{if} \quad \|\mathbf{s} - \mathbf{y}_i\| > T, \end{cases} \tag{33}$$

in which the second step holds because $\|\bar{\mathbf{y}} - \mathbf{y}_i\| \le Z \le T$. Moreover, $\sum_{i=1}^{n} \phi(\mathbf{s}, \mathbf{y}_i)$ is convex, thus $\bar{\mathbf{y}}$ is a minimizer.

(2) $\bar{\mathbf{y}}$ **is the unique minimizer.** The Hessian

$$\nabla^2 \sum_{i=1}^{n} \phi(\bar{\mathbf{y}}, \mathbf{y}_i) \ge \sum_{i=1}^{n} \mathbf{1}\left(\|\bar{\mathbf{y}} - \mathbf{y}_i\| \le T\right) = n. \tag{34}$$

Therefore $\sum_{i=1}^{n} \phi(\mathbf{s}, \mathbf{y}_i)$ is strong convex around $\bar{\mathbf{y}}$, and thus $\bar{\mathbf{y}}$ is unique. $\square$

From (30),

$$\begin{aligned}
\|g(\hat{\mu}_0(\mathcal{D}))\| &\le \sum_{i=1}^{k} \|\nabla\phi(\bar{\mathbf{y}}, \mathbf{y}_i')\| + \sum_{i=1}^{k} \|\nabla\phi(\bar{\mathbf{y}}, \mathbf{y}_i)\| \\
&\le kT + k\sum_{i=1}^{k} \|\bar{\mathbf{y}} - \mathbf{y}_i\| \\
&\le k(T + Z(\mathcal{D})). \tag{35}
\end{aligned}$$

Moreover, for $\mathbf{s}$ satisfying $\|\mathbf{s} - \bar{\mathbf{y}}\| \leq T - Z$,

$$
\begin{aligned}
\nabla g(\mathbf{s}) &= \sum_{i=1}^{k} \nabla^2 \phi(\mathbf{s}, \mathbf{y}'_i) + \sum_{i=k+1}^{n} \nabla^2 \phi(\mathbf{s}, \mathbf{y}_i) \\
&\succeq \sum_{i=k+1}^{n} \mathbf{1}(\|\mathbf{s} - \mathbf{y}_i\| \leq T) \\
&\succeq (n-k)\mathbf{I},
\end{aligned}
\tag{36}
$$

in which the last step holds because

$$
\|\mathbf{s} - \mathbf{y}_i\| \leq \|\mathbf{s} - \bar{\mathbf{y}}\| + \|\bar{\mathbf{y}} - \mathbf{y}_i\| \leq T - Z(\mathcal{D}) + Z(\mathcal{D}) = T.
\tag{37}
$$

Let $\mathbf{L}$ be a path connecting $\hat{\mu}_0(\mathcal{D})$ and $\hat{\mu}_0(\mathcal{D}')$. Then

$$
\begin{aligned}
\|g(\hat{\mu}_0(\mathcal{D})) - g(\hat{\mu}_0(\mathcal{D}'))\| &= \left\| \int_{\mathbf{L}} \nabla g(\mathbf{s}) \cdot d\mathbf{s} \right\| \\
&\geq (n-k) \min \left\{ \|\hat{\mu}_0(\mathcal{D}) - \hat{\mu}_0(\mathcal{D}')\|, T - Z(\mathcal{D}) \right\},
\end{aligned}
\tag{38}
$$

Note that from (31) and (35),

$$
\|g(\hat{\mu}_0(\mathcal{D})) - g(\hat{\mu}_0(\mathcal{D}'))\| \leq k(T + Z(\mathcal{D})).
\tag{39}
$$

From the condition $Z(\mathcal{D}) < (1 - 2k/n)T$ in Lemma 1, $(n-k)(T - Z(\mathcal{D})) > k(T + Z(\mathcal{D}))$. Hence, (38) and (39) yields

$$
\|\hat{\mu}_0(\mathcal{D}') - \hat{\mu}_0(\mathcal{D})\| \leq \frac{k(T + Z(\mathcal{D}))}{n - k}.
\tag{40}
$$

## C  Proof of Lemma 2

Denote $\mathcal{D}'$ as a dataset adjacent to $\mathcal{D}$ at user-level. Then it remains to bound $\|\hat{\mu}_0(\mathcal{D}) - \hat{\mu}_0(\mathcal{D}')\|$. Recall that in the statement of Lemma 2, we have required that there exists a dataset $\mathcal{D}^*$ such that $Z(\mathcal{D}^*) < (1 - 2(k+1)/n)T$, and $d_H(\mathcal{D}, \mathcal{D}^*) \leq k$. Denote $I$ as the set of users such that $\mathcal{D}$ and $\mathcal{D}^*$ have different values, while $\mathcal{D}$ and $\mathcal{D}'$ differ at user $u$. Now we discuss two cases.

**Case 1.** $u \notin I$. In other words, $\mathcal{D}$ and $\mathcal{D}'$ differ at a user that is the same between $\mathcal{D}$ and $\mathcal{D}^*$. In this case, $\mathcal{D}'$ has $k + 1$ users that are different from $\mathcal{D}^*$. Without loss of generality, suppose that $I = \{1, \ldots, k\}$, while $u = k + 1$. $\mathcal{D}^*$, $\mathcal{D}$ and $\mathcal{D}'$ can be written as follows:

$$
\begin{aligned}
\mathcal{D}^* &= \{D_1, \ldots, D_n\}, \tag{41} \\
\mathcal{D} &= \{D'_1, \ldots, D'_k, D_{k+1}, \ldots, D_n\}, \tag{42} \\
\mathcal{D}' &= \{D'_1, \ldots, D'_{k+1}, D_{k+2}, \ldots, D_n\}. \tag{43}
\end{aligned}
$$

For convenience of expression, denote $\mathbf{y}_i = \mathbf{y}_i(\mathcal{D})$ and $\mathbf{y}'_i = \mathbf{y}(\mathcal{D}'_i)$.

In this section, define $g_1(\mathbf{s})$ as follows:

$$
g_1(\mathbf{s}) = \sum_{i=1}^{k+1} \nabla \phi(\mathbf{s}, \mathbf{y}'_i) + \sum_{i=k+2}^{n} \nabla \phi(\mathbf{s}, \mathbf{y}_i),
\tag{44}
$$

then

$$
g_1(\hat{\mu}_0(\mathcal{D}')) = 0,
\tag{45}
$$

and

$$
\begin{aligned}
g_1(\hat{\mu}_0(\mathcal{D})) &= \sum_{i=1}^{k+1} \nabla \phi(\hat{\mu}_0(\mathcal{D}), \mathbf{y}'_i) + \sum_{i=k+2}^{n} \nabla \phi(\hat{\mu}_0(\mathcal{D}), \mathbf{y}_i) \\
&= \sum_{i=1}^{k} \nabla \phi(\hat{\mu}_0(\mathcal{D}), \mathbf{y}'_i) + \sum_{i=k+1}^{n} \nabla \phi(\hat{\mu}_0(\mathcal{D}), \mathbf{y}_i) \\
&\quad + \nabla \phi(\hat{\mu}_0(\mathcal{D}), \mathbf{y}'_{k+1}) - \nabla \phi(\hat{\mu}_0(\mathcal{D}), \mathbf{y}_{k+1}) \\
&= \nabla \phi(\hat{\mu}_0(\mathcal{D}), \mathbf{y}'_{k+1}) - \nabla \phi(\hat{\mu}_0(\mathcal{D}), \mathbf{y}_{k+1}).
\end{aligned}
\tag{46}
$$

Therefore

$$
\begin{aligned}
\|g_1(\hat{\mu}_0(\mathcal{D}))\| & \leq \left\|\nabla\phi(\hat{\mu}_0(\mathcal{D}), \mathbf{y}'_{k+1})\right\| + \left\|\nabla\phi(\hat{\mu}_0(\mathcal{D}), \mathbf{y}_{k+1})\right\| \\
& \overset{(a)}{\leq} T + \|\hat{\mu}_0(\mathcal{D}) - \mathbf{y}_{k+1}\| \\
& \overset{(b)}{\leq} T + \|\hat{\mu}_0(\mathcal{D}) - \hat{\mu}_0(\mathcal{D}^*)\| + \|\bar{\mathbf{y}}(\mathcal{D}^*) - \mathbf{y}_{k+1}\| \\
& \leq T + \omega(\mathcal{D}^*, k) + Z(\mathcal{D}^*).
\end{aligned}
\tag{47}
$$

(a) comes from (33). By taking gradient over $\mathbf{s}$, we have $\|\nabla\phi(\mathbf{s}, y)\| \leq \min\{T, \|\mathbf{s} - \mathbf{y}\|\}$. (b) comes from Lemma 6, which states that $\hat{\mu}_0(\mathcal{D}) = \bar{\mathbf{y}}$.

For all $\mathbf{s}$ satisfying $\|\mathbf{s} - \hat{\mu}_0(\mathcal{D})\| \leq T - Z(\mathcal{D}^*) - \omega(\mathcal{D}^*, k)$,

$$
\begin{aligned}
\nabla g_1(\mathbf{s}) & = \sum_{i=1}^{k+1} \nabla^2\phi(\mathbf{s}, \mathbf{y}'_i) + \sum_{i=k+2}^{n} \nabla^2\phi(\mathbf{s}, \mathbf{y}_i) \\
& \succeq \sum_{i=k+2}^{n} \mathbf{1}(\|\mathbf{s} - \mathbf{y}_i\| \leq T) \\
& \succeq (n - k - 1)\mathbf{I},
\end{aligned}
\tag{48}
$$

in which the last step holds because

$$
\begin{aligned}
\|\mathbf{s} - \mathbf{y}_i\| & \leq \|\mathbf{s} - \hat{\mu}_0(\mathcal{D})\| + \|\hat{\mu}_0(\mathcal{D}) - \bar{\mathbf{y}}\| + \|\bar{\mathbf{y}} - \mathbf{y}_i\| \\
& \leq T - Z(\mathcal{D}^*) - \omega(\mathcal{D}^*, k) + \omega(\mathcal{D}^*, k) + Z(\mathcal{D}^*) \\
& = T.
\end{aligned}
\tag{49}
$$

Then

$$
\begin{aligned}
\|g_1(\hat{\mu}_0(\mathcal{D})) - g_1(\hat{\mu}_0(\mathcal{D}'))\| & = \left\|\int_{\mathbf{L}} \nabla g_1(\mathbf{s}) d\mathbf{s}\right\| \\
& \geq (n - k - 1) \min\left\{\|\hat{\mu}_0(\mathcal{D}) - \hat{\mu}_0(\mathcal{D}')\|, T - Z(\mathcal{D}^*) - \omega(\mathcal{D}^*, k)\right\},
\end{aligned}
\tag{50}
$$

in which $\mathbf{L}$ is the line connecting $\hat{\mu}_0(\mathcal{D})$ and $\hat{\mu}_0(\mathcal{D}')$.

From (45) and (47),

$$
\|g_1(\hat{\mu}_0(\mathcal{D})) - g_1(\hat{\mu}_0(\mathcal{D}'))\| \leq T + \omega(\mathcal{D}^*, k) + Z(\mathcal{D}^*).
\tag{51}
$$

From the condition $Z(\mathcal{D}^*) < (1 - 2(k+1)/n)T$ in Lemma 2,

$$
(n - k - 1)(T - Z(\mathcal{D}^*) - \omega(\mathcal{D}^*, k)) > T + \omega(\mathcal{D}^*, k) + Z(\mathcal{D}^*).
\tag{52}
$$

Hence

$$
\|g_1(\hat{\mu}_0(\mathcal{D})) - g_1(\hat{\mu}_0(\mathcal{D}'))\| < (n - k - 1)(T - Z(\mathcal{D}^*) - \omega(\mathcal{D}^*, k)).
\tag{53}
$$

From (50) and (53), the second element in the minimum bracket in (50) will not take effect. Hence,

$$
\|\hat{\mu}_0(\mathcal{D}) - \hat{\mu}_0(\mathcal{D}')\| < T - Z(\mathcal{D}^*) - \omega(\mathcal{D}^*, k).
\tag{54}
$$

(54) will be useful in the analysis of the second case. From (50) and (53), we can also get

$$
\begin{aligned}
\|\hat{\mu}_0(\mathcal{D}) - \hat{\mu}_0(\mathcal{D}')\| & \leq \frac{T + \omega(\mathcal{D}^*, k) + Z(\mathcal{D}^*)}{n - k - 1} \\
& \leq \frac{n}{(n - k - 1)(n - k)}(T + Z(\mathcal{D}^*)),
\end{aligned}
\tag{55}
$$

in which the last step uses the bound of $\omega(\mathcal{D}^*, k)$ in Lemma 1. Now we have bounded $\|\hat{\mu}_0(\mathcal{D}) - \hat{\mu}_0(\mathcal{D}')\|$ for the first case.

**Case 2:** $u \in I$. In other words, $\mathcal{D}$ and $\mathcal{D}'$ differ in a user that is different between $\mathcal{D}$ and $\mathcal{D}^*$. Without loss of generality, suppose that $I = \{1, \ldots, k\}$, and $u = k$. $\mathcal{D}^*$, $\mathcal{D}$ and $\mathcal{D}'$ can be written as follows:

$$
\begin{aligned}
\mathcal{D}^* & = \{D_1, \ldots, D_n\}, \tag{56} \\
\mathcal{D} & = \{D'_1, \ldots, D'_k, D_{k+1}, \ldots, D_n\}, \tag{57} \\
\mathcal{D}' & = \{D'_1, \ldots, D'_{k-1}, D''_k, D_{k+1}, \ldots, D_n\}. \tag{58}
\end{aligned}
$$

In order to bound $\|\hat{\mu}_0(\mathcal{D}) - \hat{\mu}_0(\mathcal{D}')\|$, we construct a temporary dataset $\mathcal{D}_{temp}$ as follows:

$$\mathcal{D}_{temp} = \{D_1', \ldots, D_{k-1}', D_k, \ldots, D_n\}. \tag{59}$$

Define

$$g_2(\mathbf{s}) = \sum_{i=1}^{k-1} \nabla\phi(\mathbf{s}, \mathbf{y}_i') + \nabla\phi(\mathbf{s}, \mathbf{y}_k'') + \sum_{i=k+1}^{n} \nabla\phi(\mathbf{s}, \mathbf{y}_i). \tag{60}$$

Then $g_2(\hat{\mu}_0(\mathcal{D}')) = 0$, and

$$g_2(\hat{\mu}_0(\mathcal{D})) = \nabla\phi(\hat{\mu}_0(\mathcal{D}), \mathbf{y}_k'') - \nabla\phi(\hat{\mu}_0(\mathcal{D}), \mathbf{y}_k'). \tag{61}$$

From (33),

$$\|g_2(\hat{\mu}_0(\mathcal{D})) - g_2(\hat{\mu}_0(\mathcal{D}'))\| \leq 2T. \tag{62}$$

Now we bound $\|\hat{\mu}_0(\mathcal{D}) - \hat{\mu}_0(\mathcal{D}')\|$. Corresponding to (54), we get

$$\|\hat{\mu}_0(\mathcal{D}_{temp}) - \hat{\mu}_0(\mathcal{D})\| < T - Z(\mathcal{D}^*) - \omega(\mathcal{D}^*, k-1), \tag{63}$$

and

$$\|\hat{\mu}_0(\mathcal{D}_{temp}) - \hat{\mu}_0(\mathcal{D}')\| < T - Z(\mathcal{D}^*) - \omega(\mathcal{D}^*, k-1). \tag{64}$$

Denote $\mathbf{L}$ as the line connecting $\hat{\mu}_0(\mathcal{D})$ and $\hat{\mu}_0(\mathcal{D}')$. For all $\mathbf{s} \in \mathbf{L}$, $\|s - \hat{\mu}_0(\mathcal{D}_{temp})\| \leq T - Z(\mathcal{D}^*) - \omega(\mathcal{D}^*, k-1)$. Corresponding to (48), for all $\mathbf{s} \in \mathbf{L}$,

$$\nabla g_2(\mathbf{s}) \succeq (n-k)\mathbf{I}. \tag{65}$$

Therefore

$$\|g_2(\hat{\mu}_0(\mathcal{D})) - g_2(\hat{\mu}_0(\mathcal{D}'))\| \geq (n-k) \|\hat{\mu}_0(\mathcal{D}) - \hat{\mu}_0(\mathcal{D}')\|. \tag{66}$$

From (62) and (66),

$$\|\hat{\mu}_0(\mathcal{D}) - \hat{\mu}_0(\mathcal{D}')\| \leq \frac{2T}{n-k}. \tag{67}$$

Combine case 1 and 2, we get

$$\begin{aligned} LS(\mathcal{D}) &\leq \max\left\{ \frac{n(T + Z(\mathcal{D}^*))}{(n-k-1)(n-k)}, \frac{2T}{n-k} \right\} \\ &= \frac{2T}{n-k}. \end{aligned} \tag{68}$$

The last step comes from the requirement that $Z(\mathcal{D}^*) < (1 - 2(k+1)/n)T$. The proof is complete.

## D  Proof of Theorem 1

Two requirements on smooth sensitivity are shown in Definition 4, i.e.

(1) For any $\mathcal{D}$, $S(\mathcal{D}) \geq LS(\mathcal{D})$;

(2) For any neighboring $\mathcal{D}$ and $\mathcal{D}'$, $S(\mathcal{D}) \leq e^\beta S(\mathcal{D}')$.

It has been shown in [48] that based on (1) and (2), the final result $\hat{\mu}$ is $(\epsilon, \delta)$-DP. In the remainder of this section, we show that both requirements (1) and (2) are satisfied.

**For (1)**, from Lemma 1 and 2, $LS(\mathcal{D}) \leq G(\mathcal{D}, 0)$ if the corresponding conditions are satisfied. If these conditions are not satisfied, since $\|\text{Clip}(\hat{\mu}_0(\mathcal{D}), R_c)\| \leq R_c$ always hold, the sensitivity can always be bounded by $2R_c$. Hence $LS(\mathcal{D}) \leq G(\mathcal{D}, 0)$ holds for all $\mathcal{D}$. From (11), $G(\mathcal{D}, 0) \leq S(\mathcal{D})$. Therefore the requirement (1) is satisfied.

**For (2)**, it suffices to show that $G(\mathcal{D}, k) \leq G(\mathcal{D}', k+1)$. From Lemma 2, if $\Delta(\mathcal{D}')$ exists, and $k+1 \leq n/4 - 1 - \Delta(\mathcal{D}')$, then

$$G(\mathcal{D}', k+1) = \frac{2T}{n-k-1-\Delta(\mathcal{D}')}. \tag{69}$$

From the definition of $\Delta$ in (9), since $\mathcal{D}$ and $\mathcal{D}'$ are adjacent, $\Delta(\mathcal{D}) \leq \Delta(\mathcal{D}') + 1$ holds. Therefore, the condition $k + 1 \leq n/4 - 1 - \Delta(\mathcal{D}')$ yields $k \leq n/4 - 1 - \Delta(\mathcal{D})$. Use Lemma 2 again, we get

$$
\begin{aligned}
G(\mathcal{D}, k) &\leq \frac{2T}{n - k - \Delta(\mathcal{D})} \\
&\leq \frac{2T}{n - k - 1 - \Delta(\mathcal{D}')} \\
&= G(\mathcal{D}', k + 1).
\end{aligned}
\tag{70}
$$

If the conditions in Lemma 2 are not satisfied, i.e. $\Delta(\mathcal{D}')$ does not exists or $k+1 > n/4-1-\Delta(\mathcal{D}')$, then $G(\mathcal{D}', k + 1) = 2R_c$, thus $G(\mathcal{D}, k) \leq G(\mathcal{D}', k + 1)$ holds.

Now we have shown that no matter whether the conditions in Lemma 2 holds, we always have $G(\mathcal{D}, k) \leq G(\mathcal{D}', k + 1)$. From (11), it can be easily shown that $S(\mathcal{D}) \leq e^\beta S(\mathcal{D}')$.

# E  Proof of Theorem 2

In this section, we analyze the practical performance of the estimator for a random dataset.

**Notation.** Throughout this section, for convenience, we just denote $Z = Z(\mathcal{D})$, $\mathbf{Y}_i = \mathbf{y}_i(\mathcal{D})$ and $\bar{\mathbf{Y}} = \bar{\mathbf{y}}(\mathcal{D})$ for simplicity. We use capital letters here since $\mathcal{D}$ is random, thus $\mathbf{Y}_i$ and $\bar{\mathbf{Y}}$ are random variables.

From Lemma 11 in Section K, for $i = 1, \ldots, n$,

$$
P(\|\mathbf{Y}_i - \mu\| > t) \leq (d + 1)e^{-\frac{3mt^2}{32R^2}}.
\tag{71}
$$

Recall that $Z = \max_i \|\mathbf{Y}_i - \bar{\mathbf{Y}}\|$. If $\|\mathbf{Y}_i - \mu\| \leq t/2$ for all $i$, then $\|\bar{\mathbf{Y}} - \mu\| \leq t/2$ also holds, thus $Z \leq t$. Therefore

$$
\begin{aligned}
P(Z > t) &\leq \sum_{i=1}^{n} P\left(\|\mathbf{Y}_i - \mu\| > \frac{t}{2}\right) \\
&\leq n(d + 1)e^{-\frac{3mt^2}{128R^2}}.
\end{aligned}
\tag{72}
$$

Define

$$
Z_0 = \sqrt{\frac{128R^2}{3m} \ln(mn^3(d + 1))}.
\tag{73}
$$

Then

$$
P(Z > Z_0) \leq \frac{1}{mn^2}.
\tag{74}
$$

Recall that

$$
T = C_T \frac{R}{\sqrt{m}} \ln(mn^3(d + 1)),
\tag{75}
$$

with $C_T > 16\sqrt{2/3}$. Then with probability at least $1 - 1/(mn^2)$, $Z \leq T/2$ holds, thus

$$
\begin{aligned}
S(\mathcal{D}) &= \max_k e^{-\beta k} G(\mathcal{D}, k) \\
&\leq \max\left\{\max_{k \leq n/4-1} e^{-\beta k} \frac{2T}{n - k}, 2R_c e^{-\frac{1}{4}n\beta}\right\}.
\end{aligned}
\tag{76}
$$

Note that in the statement of Theorem 2, it is required that $n > (4/\beta) \ln(nR_c/T)$. This yields $2R_c e^{-n\beta/4} < 2T/n$, and $e^{-\beta k}2T/(n - k)$ decreases with $k$. Therefore

$$
S(\mathcal{D}) \leq \frac{2T}{n}.
\tag{77}
$$

It remains to bound the estimation error. The mean squared error of mean estimation can be decomposed in the following way:

$$
\begin{aligned}
\mathbb{E}\left[\|\hat{\mu}(\mathcal{D}) - \mu\|^2\right] &= \mathbb{E}\left[\|\text{Clip}(\hat{\mu}_0(\mathcal{D}), R) + \mathbf{W} - \mu\|^2\right] \\
&\stackrel{(a)}{=} \mathbb{E}\left[\|\text{Clip}(\hat{\mu}_0(\mathcal{D}), R) - \mu\|^2\right] + \mathbb{E}[\|\mathbf{W}\|^2] \\
&\leq \mathbb{E}\left[\|\hat{\mu}_0(\mathcal{D}) - \mu\|^2 \mathbf{1}(Z \leq Z_0)\right] + \mathbb{E}\left[\|\mathbf{W}\|^2 \mathbf{1}(Z \leq Z_0)\right] \\
&\quad + \mathbb{E}\left[\|\text{Clip}(\hat{\mu}_0(\mathcal{D}), R) - \mu\|^2 \mathbf{1}(Z > Z_0)\right] \\
&\quad + \mathbb{E}\left[\|\mathbf{W}\|^2 \mathbf{1}(Z > Z_0)\right] \\
&:= I_1 + I_2 + I_3 + I_4. \quad\quad (78)
\end{aligned}
$$

(a) holds because $\mathbb{E}[\mathbf{W}|D] = 0$ for any $D$, thus $\hat{\mu}_0(\mathcal{D}) - \mu$ and $\mathbf{W}$ are uncorrelated. Now we bound these four terms separately.

**Bound of $I_1$.** From Lemma 6, for $Z \leq R\ln(nd)/\sqrt{m}$, $Z < T$ holds, thus $\hat{\mu}_0(\mathcal{D}) = \bar{\mathbf{Y}}$. For convenience, denote $\mathbf{Y}$ as an i.i.d copy of $\mathbf{Y}_i$, $i = 1, \ldots, n$ and $\mathbf{X}$ as an i.i.d copy of $\mathbf{X}_{ij}$, $i = 1, \ldots, n$, $j = 1, \ldots, m$. Then

$$
\begin{aligned}
I_1 &= \mathbb{E}\left[\left\|\frac{1}{n}\sum_{i=1}^n \mathbf{Y}_i - \mu\right\|^2 \mathbf{1}\left(Z \leq \frac{R}{\sqrt{m}}\ln(nd)\right)\right] \\
&\leq \mathbb{E}\left[\left\|\frac{1}{n}\sum_{i=1}^n \mathbf{Y}_i - \mu\right\|^2\right] \\
&= \frac{1}{n}\mathbb{E}\left[\|\mathbf{Y} - \mu\|^2\right] \\
&= \frac{1}{mn}\mathbb{E}\left[\|\mathbf{X} - \mu\|^2\right] \\
&\leq \frac{R^2}{mn}. \quad\quad (79)
\end{aligned}
$$

**Bound of $I_2$.**

$$
\begin{aligned}
I_2 &\stackrel{(a)}{\leq} \frac{d}{\alpha^2}\mathbb{E}[S^2(\mathcal{D})\mathbf{1}(Z \leq Z_0)] \\
&\stackrel{(b)}{\lesssim} \frac{T^2 d}{n^2\alpha^2} \\
&\stackrel{(c)}{\lesssim} \frac{dR^2}{mn^2\epsilon^2}\ln(mn^3 d)\ln\frac{1}{\delta}. \quad\quad (80)
\end{aligned}
$$

(a) holds because $\mathbf{W} \sim \mathcal{N}(0, (\lambda^2/\alpha^2)\mathbf{I})$. (b) comes from (77). (c) comes from (3).

**Bound of $I_3$.** From (74),

$$
\begin{aligned}
I_3 &\leq 4R^2 \mathrm{P}(Z > Z_0) \\
&\leq \frac{4R^2}{mn^2}. \quad\quad (81)
\end{aligned}
$$

**Bound of $I_4$.**

$$
\begin{aligned}
I_4 &\leq \frac{d}{\alpha^2}\mathbb{E}[S^2(\mathcal{D})\mathbf{1}(Z > Z_0)] \\
&\lesssim \frac{dR^2}{\alpha^2}\mathrm{P}(Z > Z_0) \\
&\lesssim \frac{dR^2}{mn^2\epsilon^2}\ln\frac{1}{\delta}. \quad\quad (82)
\end{aligned}
$$

Now all four terms in (78) have been bounded. Therefore

$$\mathbb{E}\left[\|\hat{\mu}(\mathcal{D}) - \mu\|^2\right] \lesssim \frac{R^2}{mn} + \frac{dR^2}{mn^2\epsilon^2} \ln(mn^3 d) \ln\frac{1}{\delta}. \tag{83}$$

The proof is complete.

# F    Proof of Theorem 3

In this section, following Appendix E, we still denote $\mathbf{Y}_i = \mathbf{y}_i(\mathcal{D})$ and $\bar{\mathbf{Y}} = \bar{\mathbf{y}}(\mathcal{D})$ for simplicity.
Define

$$r_0 = \max\left\{2M_p^{\frac{1}{p}}\sqrt{\frac{1}{m}\ln\frac{3(d+1)}{\nu}}, 4M_p^{\frac{1}{p}}(3m)^{\frac{1}{p}-1}\nu^{-\frac{1}{p}}\ln\frac{3(d+1)}{\nu}\right\}, \tag{84}$$

in which $\nu = \sqrt{d}/(n\epsilon)$. From the statement of Theorem 3, $T > 4r_0$. From Lemma 13,

$$\mathrm{P}(\|\mathbf{Y}_i - \mu\| > r_0) \leq \nu = \frac{\sqrt{d}}{n\epsilon}. \tag{85}$$

Define

$$n_{out} = \sum_{i=1}^n \mathbf{1}(\|\mathbf{Y}_i - \mu\| > r_0). \tag{86}$$

Then $n_{out}$ follows a Binomial distribution with parameter $(n, p)$ with $p \leq 1/n$. $\mathbb{E}[n_{out}] \leq 1$. This indicates that with the increase of $n$, the number of outliers is still bounded by $O(1)$ with high probability.

**Lemma 7.** $\Delta(\mathcal{D}, k) \leq n_{out}$ for $k < n/4 - n_{out} - 1$.

*Proof.* From the dataset $\mathcal{D}$, we construct $\mathcal{D}^*$ as follows. Let $\mathcal{D}^* = \{D_1^*, \ldots, D_n^*\}$ such that $D_i^* = D_i$ if $\|\mathbf{Y}_i - \mu\| \leq r_0$, and $D_i^*$ is an arbitrary set with mean value $\mu$. In other words, denote $\mathbf{Y}_i^*$ as the mean value in $D_i^*$, then

$$\mathbf{Y}_i^* = \begin{cases} \mathbf{Y}_i & \text{if} \quad \|\mathbf{Y}_i - \mu\| \leq r_0 \\ \mu & \text{if} \quad \|\mathbf{Y}_i - \mu\| > r_0. \end{cases} \tag{87}$$

Then $d_H(\mathcal{D}, \mathcal{D}^*) = n_{out}$. Let $\bar{\mathbf{Y}}^* = (1/n)\sum_{i=1}^n \mathbf{Y}_i^*$. Note that

$$Z(\mathcal{D}^*) = \max_i \left\|\bar{\mathbf{Y}}^* - \mu\right\| + \max_i \|\mathbf{Y}_i^* - \mu\| \leq 2r_0 < \frac{1}{2}T. \tag{88}$$

$\square$

$G(\mathcal{D}, k)$ can be bounded using Definition 5. Since $R_c = R$, for $k \geq n/4 - n_{out} - 1$, $G(\mathcal{D}, k) \leq 2R$. Therefore, for sufficiently large $n$, if $n_{out} < n/8$, then

$$\begin{aligned} \lambda &= \max_k e^{-\beta k} G(\mathcal{D}, k) \\ &\leq \max\left\{\max_{k < n/4 - n_{out} - 1} \frac{2T}{n - k - n_{out}}, e^{-\beta(n/4 - n_{out} - 1)}\right\} \\ &\overset{(a)}{\leq} \max_{k < n/4 - n_{out} - 1} \frac{2T}{n - k - n_{out}} \\ &= \frac{8T}{3n}. \end{aligned} \tag{89}$$

(a) holds since Theorem 3 requires that $n > 8(1 + (1/\beta)\ln(n/2T))$, thus the $e^{-\beta(n/4 - n_{out} - 1)} \leq 2T/n$.

Then the mean squared error of $\hat{\mu}$ can be bounded by

$$
\begin{aligned}
\mathbb{E}\left[\|\hat{\mu}(\mathcal{D}) - \mu\|^2\right] &= \mathbb{E}\left[\|\mathrm{Clip}(\hat{\mu}_0(\mathcal{D}), R) + \mathbf{W} - \mu\|^2\right] \\
&= \mathbb{E}\left[\|\mathrm{Clip}(\hat{\mu}_0(\mathcal{D}), R) - \mu\|^2\right] + \mathbb{E}[\|\mathbf{W}\|^2] \\
&= \mathbb{E}\left[\|\mathrm{Clip}(\hat{\mu}_0(\mathcal{D}), R) - \mu\|^2 \, \mathbf{1}(n_{out} < \tfrac{1}{8}n)\right] + \mathbb{E}\left[\|\mathbf{W}\|^2 \, \mathbf{1}(n_{out} < \tfrac{1}{8}n)\right] \\
&\quad + \mathbb{E}\left[\|\mathrm{Clip}(\hat{\mu}_0(\mathcal{D}), R) - \mu\|^2 \, \mathbf{1}(n_{out} \ge \tfrac{1}{8}n)\right] + \mathbb{E}\left[\|\mathbf{W}\|^2 \, \mathbf{1}(n_{out} \ge \tfrac{1}{8}n)\right] \\
&:= I_1 + I_2 + I_3 + I_4. \tag{90}
\end{aligned}
$$

From Chernoff inequality,

$$
\mathrm{P}(n_{out} > l) \le e^{-n\nu}\left(\frac{en\nu}{l}\right)^l. \tag{91}
$$

Thus

$$
\mathrm{P}\left(n_{out} > \frac{1}{8}n\right) \le \left(\frac{8e\sqrt{d}}{n\epsilon}\right)^{\frac{n}{8}}, \tag{92}
$$

which decays faster than any polynomial. Therefore, $I_3$ and $I_4$ can be neglected in asymptotic analysis. Now we bound $I_1$ and $I_2$.

**Bound of $I_1$.** Note that

$$
\|\hat{\mu}_0(\mathcal{D}) - \mu\| \le \|\hat{\mu}_0(\mathcal{D}) - \hat{\mu}_0(\mathcal{D}^*)\| + \|\hat{\mu}_0(\mathcal{D}^* - \mu)\|. \tag{93}
$$

Since $Z(\mathcal{D}^*) < T$, $\hat{\mu}_0(\mathcal{D}^*) = \bar{\mathbf{Y}}^*$, then

$$
\begin{aligned}
\mathbb{E}\left[\|\hat{\mu}_0(\mathcal{D}^*) - \mu\|^2\right] &= \mathbb{E}\left[\|\bar{\mathbf{Y}}^* - \mu\|^2\right] \\
&\le \mathrm{tr}(\mathrm{Var}[\bar{\mathbf{Y}}^*]) + \left\|\mathbb{E}[\bar{\mathbf{Y}}^*] - \mu\right\|^2 \\
&\lesssim \frac{1}{mn} + r_0\nu, \tag{94}
\end{aligned}
$$

in which the last step uses Lemma 14.

From Lemma 1,

$$
\begin{aligned}
\|\hat{\mu}_0(\mathcal{D}) - \hat{\mu}_0(\mathcal{D}^*)\| &\le \omega(\mathcal{D}^*, n_{out}) \\
&\le \frac{n_{out}(T + Z(\mathcal{D}^*))}{n - n_{out}} \\
&\le \frac{\frac{3}{2}T n_{out}}{n - n_{out}}. \tag{95}
\end{aligned}
$$

The expectation can be bounded by

$$
\begin{aligned}
\mathbb{E}\left[\|\hat{\mu}_0(\mathcal{D}) - \hat{\mu}_0(\mathcal{D}^*)\|^2 \, \mathbf{1}\left(n_{out} < \frac{n}{8}\right)\right] &\le \mathbb{E}\left[\left(\frac{\frac{3}{2}T n_{out}}{n - \frac{n}{8}}\right)^2\right] \\
&\sim \frac{T^2}{n^2}\mathbb{E}[n_{out}^2] \\
&\sim \frac{T^2}{n^2}. \tag{96}
\end{aligned}
$$

Therefore

$$
I_1 \lesssim \frac{T^2}{n^2} + \frac{1}{mn} + \nu^2 r_0^2. \tag{97}
$$

**Bound of $I_2$.**

$$\mathbb{E}\left[\|\mathbf{W}\|^2 \, \mathbf{1}\left(n_{out} < \frac{n}{8}\right)\right] \leq \frac{d}{\alpha^2}\mathbb{E}\left[\lambda^2 \mathbf{1}(n_{out} < \frac{n}{8})\right] \lesssim \frac{dT^2}{n^2\epsilon^2}\ln\frac{1}{\delta}, \tag{98}$$

in which the last step uses (3) and (89).

Combine all terms, the final bound on the mean squared error is

$$
\begin{aligned}
\mathbb{E}\left[\|\hat{\mu}(\mathcal{D}) - \mu\|^2\right] &\lesssim &\frac{1}{mn} + \frac{dT^2}{n^2\epsilon^2}\ln\frac{1}{\delta} + r_0^2\nu^2 \\
&\sim &\frac{1}{mn} + \frac{dT^2}{n^2\epsilon^2}\ln\frac{1}{\delta} \\
&\sim &\frac{1}{mn} + \frac{d}{n^2\epsilon^2}\ln\frac{1}{\delta}\left(\frac{\ln(nd)}{m} + m^{\frac{2}{p}-2}n^{\frac{2}{p}}\epsilon^{\frac{2}{p}}d^{-\frac{1}{p}}\ln^2(nd)\right), &(99)
\end{aligned}
$$

in which the second step uses $T > 4r_0$ (from (13) and (84)) and $\nu = \sqrt{d}/(n\epsilon)$. The proof is complete.

## G   Comment on Two-stage Approach

In this section, we provide a brief comment on the two-stage approach WME in [34].

*1) For heavy-tailed distributions.* We follow the steps of Appendix D.4 in [34].

[34] defined a $(\tau, \gamma)$ concentration, which requires that there exists a point $\mathbf{c}$, with probability $1 - \gamma$, $\|\mathbf{Y}_i - \mathbf{c}\| \leq \tau$ for all $i$, in which $\mathbf{Y}_i$ is the $i$-th user-wise average. We use capital letter here to denote that it is a random variable.

From Appendix D.4 in [34], $\gamma \sim 1/(mn^2\epsilon^2)$ is used. From Lemma 13, let $\nu = 1/(mn^3\epsilon^2)$, we have

$$\|\mathbf{Y} - \mu\| \leq \max\left\{2M_p^{\frac{1}{p}}\sqrt{\frac{1}{m}\ln\frac{3(d+1)}{\nu}}, 4M_p^{\frac{1}{p}}(3m)^{\frac{1}{p}-1}\nu^{-\frac{1}{p}}\ln\frac{3(d+1)}{\nu}\right\} \tag{100}$$

with probability at least $1 - \nu$, in which $\mathbf{Y}$ is i.i.d with $\mathbf{Y}_i$, $i = 1, \ldots, n$. The value of $\tau$ can be obtained by taking union bound for all $i$:

$$\tau \sim \sqrt{\frac{1}{m}\ln(mn^3d)} + m^{\frac{1}{p}-1}(mn^3)^{\frac{1}{p}}\ln(mn^3d). \tag{101}$$

Follow other parts of Appendix D.4 in [34], we can get

$$\mathbb{E}[\|\hat{\mu} - \mu\|^2] \lesssim \frac{1}{mn} + \frac{d\ln^2(mnd)\ln\frac{1}{\delta}}{n^2\epsilon^2}\left[\frac{1}{m} + m^{\frac{4}{p}-2}n^{\frac{6}{p}}\ln(nmd)\right]. \tag{102}$$

Moreover, [34] has already shown the tightness of the mean squared error for the bounded case. For heavy-tailed distributions, following Appendix D.4 in [34], it can also be shown that the bound (102) is tight.

*2) For imbalanced users.* For simplicity, we focus on the one dimensional problem. With the two-stage approach, for users with different $m_i$, the final estimate will be

$$\hat{\mu} = \frac{1}{N}\sum_{i=1}^{n}m_i\Pi_{[a,b]}Y_i, \tag{103}$$

in which $\pi_{[a,b]}$ means clipping on $[a, b]$. The sensitivity is determined by the user with maximum $m_i$. Let $m_{\max} = \max_i m_i$. Then the sensitivity if $m_{\max}(b - a)/n$.

From [34], now suppose that $m_i$ are the same for all $i$ except one that is significantly larger. Then $\tau \sim R\sqrt{n\ln n/N}$, $b - a = 4\tau$, the sensitivity scales as $(Rm_{\max}/N)\sqrt{n\ln n/N}$. Denote $\gamma_0 = nm_{\max}/N$ as the ratio between maximum $m_i$ and average $m_i$. Then the mean squared error induced by privacy mechanism is

$$\mathbb{E}[W^2] \gtrsim \frac{R^2m_{\max}^2}{N^2}\frac{n\ln N}{N} = \frac{\gamma_0^2R^2\ln n}{Nn\epsilon^2}. \tag{104}$$

# H    Proof of Lemma 3

Following the proof of Theorem 2 in Appendix E, denote $\mathbf{y}_i = \mathbf{y}_i(\mathcal{D})$, $\mathbf{y}'_i = \mathbf{y}_i(\mathcal{D}')$ and $\bar{\mathbf{y}} = (1/n)\sum_{i=1}^{n}\mathbf{y}_i$. The proof begins with the following lemma.

**Lemma 8.** *If $T_i > Z_i(\mathcal{D})$ for all $i$, then $\hat{\mu}_0(\mathcal{D}) = \bar{\mathbf{y}}$.*

*Proof.* We just need to prove that $\bar{\mathbf{y}}$ is a minimizer of $\sum_{i=1}^{n} w_i\phi_i(\mathbf{s}, \mathbf{y}_i)$. Since $Z_i < T_i$ for all $i$,

$$\nabla\phi_i(\bar{\mathbf{y}}, \mathbf{y}_i) = \bar{\mathbf{y}} - \mathbf{y}_i. \tag{105}$$

Then

$$\sum_{i=1}^{n} w_i\nabla\phi_i(\bar{\mathbf{y}}, \mathbf{y}_i) = \sum_{i=1}^{n} w_i(\bar{\mathbf{y}} - \mathbf{y}_i) = 0. \tag{106}$$

$\square$

Now we derive the sensitivity. Without loss of generality, we assume that first $k$ users are replaced by others. Define

$$g(\mathbf{s}) = \sum_{i\in I} w_i\nabla\phi(\mathbf{s}, \mathbf{y}'_i) + \sum_{i\in[n]\setminus I} w_i\nabla\phi(\mathbf{s}, \mathbf{y}_i), \tag{107}$$

then $g(\hat{\mu}_0(\mathcal{D}')) = 0$, and

$$
\begin{aligned}
g(\hat{\mu}_0(\mathcal{D})) &= \sum_{i\in I} w_i\nabla\phi(\hat{\mu}_0(\mathcal{D}), \mathbf{y}'_i) + \sum_{i\in[n]\setminus I} w_i\nabla\phi(\hat{\mu}_0(\mathcal{D}), \mathbf{y}_i) \\
&= \sum_{i\in I} w_i\nabla\phi(\hat{\mu}_0(\mathcal{D}), \mathbf{y}'_i) - \sum_{i\in I} w_i\nabla\phi(\hat{\mu}_0(\mathcal{D}), \mathbf{y}_i) + \sum_{i=1}^{n} w_i\nabla\phi(\hat{\mu}_0(\mathcal{D}), \mathbf{y}_i) \\
&= \sum_{i\in I} w_i\nabla\phi(\hat{\mu}_0(\mathcal{D}), \mathbf{y}'_i) - \sum_{i\in I} w_i\nabla\phi(\hat{\mu}_0(\mathcal{D}), \mathbf{y}_i). 
\end{aligned}
\tag{108}
$$

Recall that $Z = \max_i \|\bar{\mathbf{y}} - \mathbf{y}_i\|$. From Lemma 8, if $Z_i < T_i$ for all $i$, then

$$\|\hat{\mu}_0(\mathcal{D}) - \mathbf{y}_i\| = \|\bar{\mathbf{y}} - \mathbf{y}_i\| \leq Z, \tag{109}$$

thus

$$\|g(\hat{\mu}_0(\mathcal{D}))\| \leq \sum_{i\in I} w_i(T_i + Z_i). \tag{110}$$

From (107),

$$
\begin{aligned}
\nabla g(\mathbf{s}) &= \sum_{i\in I} w_i\nabla^2\phi(\mathbf{s}, \mathbf{y}'_i) + \sum_{i\in[n]\setminus I} w_i\nabla^2\phi(\mathbf{s}, \mathbf{y}_i) \\
&\succeq \sum_{i\in[n]\setminus I} w_i\mathbf{1}(\|\mathbf{s} - \mathbf{y}_i\| \leq T_i).
\end{aligned}
\tag{111}
$$

For all $\mathbf{s}$ satisfying $\|\mathbf{s} - \bar{\mathbf{y}}\| \leq \min_i(T_i - Z_i)$, $\|\mathbf{s} - \mathbf{y}_i\| \leq T_i$ always holds, thus

$$\nabla g(\mathbf{s}) \succeq \left(\sum_{i\in[n]\setminus I} w_i\right)\mathbf{I} = \left(N - \sum_{i\in I} w_i\right)\mathbf{I}. \tag{112}$$

Under condition $h(\mathcal{D}, 1) \leq \min_i(T_i - Z_i(\mathcal{D}))$,

$$\|\hat{\mu}_0(\mathcal{D}') - \hat{\mu}_0(\mathcal{D})\| \leq \frac{\sum_{i\in I} w_i(T_i + Z_i(\mathcal{D}))}{N - \sum_{i\in I} w_i}. \tag{113}$$

Recall that we have assumed $\mathcal{D}' = \{D'_1, \ldots, D'_k, D_{k+1}, \ldots, D_n\}$, i.e. first $k$ users are replaced by others. Actually, the replaced users can be arbitrarily selected from $n$ users. Therefore, a simple generalization yields:

$$\omega(\mathcal{D}, k) \leq \max_{I\subseteq[n], |I|=k} \frac{\sum_{i\in I} w_i(T_i + Z_i(\mathcal{D}))}{N - \sum_{i\in I} w_i}. \tag{114}$$

The proof is complete.

# I  Proof of Lemma 4

Similar to the proof of Lemma 2 in Appendix C, we analyze two cases.

For the first case, $u \notin I$, in which $u$ and $I$ have the same definition as in Appendix 5, without loss of generality, suppose $\mathcal{D}$ and $\mathcal{D}^*$ differ in the first $k$ users, while $\mathcal{D}$ and $\mathcal{D}'$ differ in the $(k+1)$-th user. Then

$$\|\hat{\mu}_0(\mathcal{D}') - \hat{\mu}_0(\mathcal{D})\| \leq \frac{m_{k+1}(T_{k+1} + Z_{k+1}(\mathcal{D}^*) + \omega(\mathcal{D}^*, k))}{N - \sum_{i=1}^{k+1} w_i}. \tag{115}$$

From Lemma 3 and the condition $h(\mathcal{D}^*, k+1) < \min_i(T_i - Z_i(\mathcal{D}^*))$ in Lemma 4,

$$\omega(\mathcal{D}^*, k) \leq h(\mathcal{D}^*, k) < \min_i(T_i - Z_i(\mathcal{D}^*)) \leq T_{k+1} - Z_{k+1}(\mathcal{D}^*), \tag{116}$$

thus

$$\|\hat{\mu}_0(\mathcal{D}') - \hat{\mu}_0(\mathcal{D})\| \leq \frac{2m_{k+1}T_{k+1}}{N - \sum_{i=1}^{k+1} w_i}. \tag{117}$$

For the second case, following steps in Appendix B,

$$\|\hat{\mu}_0(\mathcal{D}') - \hat{\mu}_0(\mathcal{D})\| \leq \frac{2m_k T_k}{N - \sum_{i=1}^{k} w_i} \tag{118}$$

Taking maximum, we have

$$LS(\mathcal{D}) \leq \frac{2\max\limits_{i \in [n]} w_i T_i}{N\left(1 - \gamma(k+1)\right)}. \tag{119}$$

# J  Proof of Theorem 5

Similar to the proof of Theorem 2, denote $Z_i = Z_i(\mathcal{D})$, $\mathbf{Y}_i = \mathbf{y}_i(\mathcal{D})$ and $\bar{\mathbf{Y}} = (1/n)\sum_{i=1}^n \mathbf{Y}_i$. Denote

$$N_c = \sum_{i=1}^{n} m_i \wedge m_c, \tag{120}$$

then from the statement in Theorem 5, $w_i = m_i \wedge m_c / N_c$. From the statement of Theorem 5, recall that $m_c = \gamma N/n$.

The proof starts with the following lemma.

**Lemma 9.** *With probability $1 - (n+1)/(Nn^2)$, for all $i$,*

$$Z_i \leq \sqrt{\frac{32}{3} R^2 \ln(Nn^2(d+1))} \left(\frac{1}{N_c} + \frac{1}{\sqrt{m_i}}\right). \tag{121}$$

From now on, denote $a_i$ as the right hand side of (121).

*Proof.* From Lemma 11,

$$P(\|\bar{\mathbf{Y}} - \mu\| > t) \leq (d+1)e^{-\frac{3Nt^2}{32R^2}}, \tag{122}$$

and

$$P(\|\mathbf{Y}_i - \mu\| > t) \leq (d+1)e^{-\frac{3m_i t^2}{32R^2}}. \tag{123}$$

Recall that $Z_i = \|\bar{\mathbf{Y}} - \mathbf{Y}_i\|$. Therefore

$$\begin{aligned}
P\left(\cup_{i=1}^n \{Z_i > a_i\}\right) &\leq P\left(\|\bar{\mathbf{Y}} - \mu\| > \sqrt{\frac{32R^2}{3N} \ln(Nn^2(d+1))}\right) \\
&\quad + \sum_{i=1}^n P\left(\|\mathbf{Y}_i - \mu\| > \sqrt{\frac{32R^2}{3m_i} \ln(Nn^2(d+1))}\right) \\
&\leq \frac{1}{Nn^2} + n\frac{1}{Nn^2} \\
&= \frac{n+1}{Nn^2}.
\end{aligned} \tag{124}$$

The proof of Lemma 9 is complete. ☐

Since $Z_i \leq a_i < T_i$ for all $i$, from Lemma 8, $\hat{\mu}_0(\mathcal{D}) = \bar{\mathbf{Y}}$. Moreover, we show that $\Delta(\mathcal{D}) = 0$, in which $\Delta$ is defined in (17). This needs the following lemma.

**Lemma 10.** *For $k \leq n/(8\gamma)$, if $Z_i \leq a_i$ for all $i$, in which $a_i$ is the right hand side of* (121)*, then*

$$h(\mathcal{D}, k) < \min_{i \in [n]}(T_i - Z_i). \tag{125}$$

*Proof.* Recall $T_i$ in (19), let

$$A = C_T R \sqrt{\ln(Nn^2(d+1))}, \tag{126}$$

then $T_i = A/\sqrt{m_i \wedge m_c}$, thus

$$
\begin{aligned}
\min_i(T_i - Z_i) &\geq \min_i \left( \frac{A}{\sqrt{m_i \wedge m_c}} - a_i \right) \\
&\geq \min_i \left( \frac{A}{\sqrt{m_i \wedge m_c}} - 2\sqrt{\frac{32R^2}{3m_i} \ln(Nn^2(d+1))} \right) \\
&\geq \min_i \left( \frac{A}{\sqrt{m_i \wedge m_c}} - \frac{A}{2\sqrt{m_i}} \right) \\
&\geq \frac{A}{2\sqrt{m_c}} \\
&= \frac{A}{2}\sqrt{\frac{n}{\gamma N}}. 
\end{aligned}
\tag{127}
$$

Now we provide an upper bound of $h(\mathcal{D}, k)$:

$$
\begin{aligned}
h(\mathcal{D}, k) &\overset{(a)}{\leq} \frac{\sum_{i=n-k+1}^{n} w_i(T_i + a_i)}{\sum_{i=1}^{n-k} w_i} \\
&= \frac{\sum_{i=n-k-1}^{n}(m_i \wedge m_c)(T_i + a_i)}{\sum_{i=1}^{n-k} m_i \wedge m_c} \\
&\leq \frac{\sum_{i=n-k+1}^{n}(m_i \wedge m_c)\left( \frac{A}{\sqrt{m_i \wedge m_c}} + \frac{A}{2\sqrt{m_i}} \right)}{N_c - km_c} \\
&\leq \frac{\frac{3}{2}kA\sqrt{m_c}}{\frac{1}{2}N - \frac{k\gamma N}{n}} \\
&\overset{(b)}{<} \frac{\frac{3n}{8\gamma}A\sqrt{\frac{\gamma N}{n}}}{\frac{3}{4}N} \\
&= \frac{A}{2}\sqrt{\frac{n}{\gamma N}}. 
\end{aligned}
\tag{128}
$$

(a) comes from (16), and that $Z_i \leq a_i$ for all $i$. (b) uses the condition $k < n/(8\gamma)$.

Combine (127) and (128), (125) holds. The proof of Lemma 10 is complete. ☐

Now it remains to bound of $G(\mathcal{D}, k)$. From Lemma 10, if $Z_i \leq a_i$ for all $i$, then $\Delta(\mathcal{D}) = 0$, since $h(\mathcal{D}^*, k_0) < \min_i(T_i - Z_i)$. Then for all $k \leq k_0 - 1$,

$$
\begin{aligned}
G(\mathcal{D}, k) & \overset{(a)}{\leq} \frac{2 \max_i w_i T_i}{\sum_{i=1}^{n-k-1} w_i} \\
& = 2 \frac{\max_i (m_i \wedge m_c) T_i}{\sum_{i=1}^{n-k-1} m_i \wedge m_c} \\
& \leq 2 \frac{A\sqrt{m_c}}{N_c - (k+1) m_c} \\
& \overset{(b)}{\leq} \frac{2A\sqrt{m_c}}{\frac{1}{2} N - (k+1)\frac{\gamma N}{n}} \\
& = \frac{4 A m_c}{N \left(1 - 2\gamma \frac{k+1}{n}\right)} \\
& \overset{(c)}{\leq} \frac{16A}{3N} \sqrt{m_c} \\
& = \frac{16A}{3} \sqrt{\frac{\gamma}{Nn}}.
\end{aligned}
\tag{129}
$$

(a) comes from Definition 6. Note that $\Delta(\mathcal{D}) = 0$. For $k \leq k_0 - 1$, $G(\mathcal{D}, k)$ is $h(\mathcal{D}, 1)$ if $h(\mathcal{D}, 1) \leq \min_i (T_i - Z_i(\mathcal{D}))$ holds, or $2 \max_i w_i T_i / (\sum_{i=1}^{n-k-1} w_i)$. It can be shown that the former one is less than the latter, thus (a) holds. For (b), from Assumption 3,

$$
N_c = \sum_{i=1}^{n} m_i \wedge m_c \geq N - \sum_{k: m_k > \gamma N/n} m_k \geq N - \frac{N}{2} = \frac{N}{2}.
\tag{130}
$$

(c) holds because $\gamma(k+1)/n \leq \gamma k_0/n \leq 1/8$.

From (129), the smooth sensitivity can be bounded by

$$
S(\mathcal{D}) \leq \max \left\{ \frac{16A}{3} \sqrt{\frac{\gamma}{Nn}}, 2Re^{-\beta k_0} \right\}.
\tag{131}
$$

Recall that $k_0 = \lfloor n/(8\gamma) \rfloor$. In Theorem 5, it is required that $n > 8\gamma(1 + (1/2\beta) \ln(Nn))$, thus $k_0 \geq \ln(Nn)/(2\beta)$, and $e^{-\beta k_0} = 1/\sqrt{Nn}$. Therefore, the second term in (131) does not dominate. This result indicates that as long as $Z_i \leq a_i$ for all $i$,

$$
S(\mathcal{D}) \leq \frac{16A}{3} \sqrt{\frac{\gamma}{Nn}}.
\tag{132}
$$

Now we bound the mean squared error. Denote $E$ as the event such that $Z_i \leq a_i$ for all $i$. Then

$$
\begin{aligned}
\mathbb{E}\left[\|\hat{\mu}(D) - \mu\|^2\right] & \leq \mathbb{E}\left[\|\hat{\mu}_0(D) - \mu\|^2 \mathbf{1}(E)\right] + \mathbb{E}\left[\|W\|^2 \mathbf{1}(E)\right] \\
& \quad + \mathbb{E}\left[\|\text{Clip}(\hat{\mu}_0(D), R) - \mu\|^2 \mathbf{1}(E^c)\right] + \mathbb{E}\left[\|W\|^2 \mathbf{1}(E^c)\right] \\
& := I_1 + I_2 + I_3 + I_4.
\end{aligned}
\tag{133}
$$

**Bound of $I_1$.**

$$
\begin{aligned}
I_1 &= \mathbb{E}\left[\left\|\bar{\mathbf{Y}} - \mu\right\|^2 \mathbf{1}(E)\right] \\
&\leq \mathbb{E}\left[\left\|\bar{\mathbf{Y}} - \mu\right\|^2\right] \\
&= \operatorname{tr} \operatorname{Var}\left[\sum_i w_i \mathbf{Y}_i\right] \\
&= \sum_i w_i^2 \frac{R^2}{m_i} \\
&= \frac{\sum_i (m_i \wedge m_c)^2 \frac{R^2}{m_i}}{\left(\sum_i m_i \wedge m_c\right)^2} \\
&\leq \frac{1}{\sum_i (m_i \wedge m_c)} \\
&= \frac{1}{N_c} \\
&\leq \frac{2}{N}.
\end{aligned}
\tag{134}
$$

**Bound of $I_2$.**

$$
\begin{aligned}
I_2 &= \frac{\mathbb{E}[S^2(\mathcal{D})\mathbf{1}(E)]}{\alpha^2} d \\
&\lesssim \frac{d}{\alpha^2} A^2 \frac{\gamma}{Nn} \\
&\sim \frac{dR^2\gamma}{Nn\epsilon^2} \ln(Nn^2 d) \ln\frac{1}{\delta}.
\end{aligned}
\tag{135}
$$

**Bound of $I_3$.**

$$
\begin{aligned}
I_3 &\leq 4R^2 \mathrm{P}(E^c) \\
&\leq 4\frac{n+1}{Nn^2}.
\end{aligned}
\tag{136}
$$

**Bound of $I_4$.**

$$
I_4 \lesssim \frac{\mathbb{E}[\lambda^2 \mathbf{1}(E^c)]}{\alpha^2} d \lesssim \frac{dR^2}{\epsilon^2} \ln\frac{1}{\delta}\frac{1}{Nn}.
\tag{137}
$$

$I_3$ and $I_4$ converges to zero faster than any polynomial. Therefore

$$
\mathbb{E}\left[\left\|\hat{\mu}(D) - \mu\right\|^2\right] \lesssim \frac{R^2}{mn} + \frac{dR^2\gamma}{N\epsilon^2} \ln(Nn^2 d) \ln\frac{1}{\delta}.
\tag{138}
$$

## K  Common Lemmas

**Lemma 11.** *(Concentration inequality of bounded random vector) Given a random vector $\mathbf{X}$ supported at $B_d(\mathbf{0}, R)$, and $\mathbb{E}[\mathbf{X}] = \mu$. $\mathbf{X}_1, \ldots, \mathbf{X}_m$ are $m$ i.i.d copies of $\mathbf{X}$. Denote $\bar{\mathbf{X}}$ as the sample mean, i.e. $\bar{\mathbf{X}} = (1/m)\sum_{j=1}^m \mathbf{X}_j$. Then*

$$
P(\left\|\bar{\mathbf{X}} - \mu\right\| > t) \leq (d+1)e^{-\frac{3mt^2}{32R^2}}.
\tag{139}
$$

*Proof.* We use the following lemma.

**Lemma 12.** *( [88], Lemma 1.6.2) Let $\mathbf{U}_1, \ldots, \mathbf{U}_m$ be independent centered random vectors with dimension $d$. Assume that each one is uniformly bounded, i.e. for $j = 1, \ldots, m$,*

$$\mathbb{E}[\mathbf{U}_j] = 0, \tag{140}$$

*and with probability* $1$,

$$\|\mathbf{U}_j\| \leq L. \tag{141}$$

*Let $\mathbf{Z} = \sum_{j=1}^{m} \mathbf{U}_j{}^2$, and define*

$$\gamma(\mathbf{Z}) = \left| \sum_{j=1}^{m} \mathbb{E}[\mathbf{U}_j^T \mathbf{U}_j] \right|. \tag{142}$$

*Then for all $t > 0$,*

$$P(\|\mathbf{Z}\| > t) \leq (d+1) \exp \left[ -\frac{t^2/2}{\gamma(\mathbf{Z}) + Lt/3} \right]. \tag{143}$$

Now we prove Lemma 11 based on Lemma 12. Let $\mathbf{U}_j = \mathbf{X}_j - \mu$. Since $\|\mathbf{X}_j\| \leq R$, $\|\mu\| \leq R$ holds, $\|\mathbf{X}_j - \mu\| \leq 2R$ always holds, and $\gamma(\mathbf{Z}) = 4mR^2$. Hence

$$P(\|\mathbf{Z}\| > t) \leq (d+1) \exp \left[ -\frac{t^2/2}{4mR^2 + \frac{2}{3}Rt} \right]. \tag{144}$$

Hence

$$
\begin{aligned}
P\left(\|\bar{\mathbf{X}} - \mu\| > t\right) &\leq (d+1) \exp \left[ -\frac{m^2 t^2}{8mR^2 + \frac{4}{3}Rtm} \right] \\
&\leq (d+1) \exp \left[ -\frac{3mt^2}{32R^2} \right].
\end{aligned} \tag{145}
$$

$\square$

**Lemma 13.** *(Concentrated inequality of unbounded random vector) Given a random vector $\mathbf{X}$, $\mathbb{E}[\mathbf{X}] = \mu$, and $\mathbb{E}\left[ \|\mathbf{X} - \mu\|^p \right] \leq M_p$. $\mathbf{X}_1, \ldots, \mathbf{X}_m$ are $m$ i.i.d copies of $\mathbf{X}$. Denote $\bar{\mathbf{X}}$ as the sample mean. Then with probability at least $1 - \nu$,*

$$\left\| \bar{\mathbf{X}} - \mu \right\| \leq \max \left\{ 2M_p^{\frac{1}{p}} \sqrt{\frac{1}{m} \ln \frac{3(d+1)}{\nu}}, \, 4M_p^{\frac{1}{p}} (3m)^{\frac{1}{p}-1} \nu^{-\frac{1}{p}} \ln \frac{3(d+1)}{\nu} \right\}. \tag{146}$$

*Proof.* We still use Lemma 12. Pick $r > 0$, whose exact value will be determined later. Define

$$\mathbf{U}_j := (\mathbf{X}_j - \mu) \mathbf{1}(\|\mathbf{X}_j - \mu\| \leq r), \tag{147}$$

and

$$\mathbf{V}_j := (\mathbf{X}_j - \mu) \mathbf{1}(\|\mathbf{X}_j - \mu\| > r). \tag{148}$$

Then

$$\bar{\mathbf{X}} - \mu = \frac{1}{m} \sum_{j=1}^{m} \mathbf{U}_j + \frac{1}{m} \sum_{j=1}^{m} \mathbf{V}_j, \tag{149}$$

and

$$P\left(\|\bar{\mathbf{X}} - \mu\| > t\right) \leq P\left(\left\| \frac{1}{m} \sum_{j=1}^{m} \mathbf{U}_j \right\| > t\right) + P\left(\left\| \frac{1}{m} \sum_{j=1}^{m} \mathbf{V}_j \right\| > 0\right). \tag{150}$$

---

[2]This definition of $\mathbf{Z}$ is only used in this section.

Let $\mathbf{Z} = \sum_{j=1}^m \mathbf{U}_j$. Then from (142),

$$\gamma(\mathbf{Z}) = \mathbb{E}\left[\sum_{j=1}^m \mathbf{U}_j^T \mathbf{U}_j\right] \leq m\mathbb{E}[\|\mathbf{X} - \mu\|^2] \leq mM_p^{\frac{2}{p}}. \tag{151}$$

Note that $\|\mathbf{U}_j\| < r$ always holds. Therefore, from Lemma 12,

$$
\begin{aligned}
\mathbf{P}\left(\|\mathbf{X}\| > t\right) &\leq& (d+1)\exp\left[-\frac{t^2/2}{mM_p^{\frac{2}{p}} + \frac{1}{3}rt}\right] \\
&\leq& (d+1)\exp\left[-\min\left\{\frac{t^2}{4mM_p^{\frac{2}{p}}}, \frac{3t}{4r}\right\}\right],
\end{aligned}
\tag{152}
$$

and

$$
\begin{aligned}
\mathbf{P}\left(\left\|\frac{1}{m}\sum_{j=1}^m \mathbf{U}_j\right\| > t\right) &=& \mathbf{P}(\|\mathbf{Z}\| > mt) \\
&\leq& (d+1)\exp\left[-\min\left\{\frac{mt^2}{4M_p^{\frac{2}{p}}}, \frac{3mt}{4r}\right\}\right] \\
&\leq& (d+1)e^{-\frac{mt^2}{4M_p^{\frac{2}{p}}}} + (d+1)e^{-\frac{3mt}{4r}}.
\end{aligned}
\tag{153}
$$

Now we have bounded the first term in (150). For the second term in (150),

$$
\begin{aligned}
\mathbf{P}\left(\left\|\frac{1}{m}\sum_{j=1}^m \mathbf{V}_j\right\| > 0\right) &\leq& \mathbf{P}\left(\cup_{j=1}^m \{\|\mathbf{V}_j\| > 0\}\right) \\
&\leq& m\mathbf{P}\left(\|\mathbf{X} - \mu\| > r\right) \\
&\leq& mM_p r^{-p}.
\end{aligned}
\tag{154}
$$

Therefore, from (150), (153) and (154),

$$\mathbf{P}\left(\|\bar{\mathbf{X}} - \mu\| > t\right) \leq (d+1)e^{-\frac{mt^2}{4M_p^{\frac{2}{p}}}} + (d+1)e^{-\frac{3mt}{4r}} + M_p m r^{-p}. \tag{155}$$

To make the right hand side of (155) to be no more than $\nu$, we let each term to be no more than $\nu/3$. Note that now $r$ has not be determined. Therefore, we let the third term equals $\nu/3$ first, thus

$$r = \left(\frac{3M_p m}{\nu}\right)^{\frac{1}{p}}. \tag{156}$$

Then we let

$$t = \max\left\{2M_p^{\frac{1}{p}}\sqrt{\frac{1}{m}\ln\frac{3(d+1)}{\nu}}, 4M_p^{\frac{1}{p}}(3m)^{\frac{1}{p}-1}\nu^{-\frac{1}{p}}\ln\frac{3(d+1)}{\nu}\right\}. \tag{157}$$

With (156) and (157), all three terms in the right hand side of (155) will be no more than $\nu/3$. Therefore, with probability at least $1 - \nu$,

$$\|\bar{\mathbf{X}} - \mu\| \leq \max\left\{2M_p^{\frac{1}{p}}\sqrt{\frac{1}{m}\ln\frac{3(d+1)}{\nu}}, 4M_p^{\frac{1}{p}}(3m)^{\frac{1}{p}-1}\nu^{-\frac{1}{p}}\ln\frac{3(d+1)}{\nu}\right\}. \tag{158}$$

The proof is complete. $\qquad\square$

**Lemma 14.** *(Bias caused by outliers) Given a random vector $\mathbf{X}$, $\mathbb{E}[\mathbf{X}] = \mu$, and $\mathbb{E}\left[\|\mathbf{X} - \mu\|^p\right] \le M_p$. $\mathbf{X}_1, \ldots, \mathbf{X}_m$ are $m$ i.i.d copies of $\mathbf{X}$. Denote $\bar{\mathbf{X}}$ as the sample mean, and*

$$\bar{\mathbf{X}}^* = \begin{cases} \bar{\mathbf{X}} & \text{if} \quad \left\|\bar{\mathbf{X}} - \mu\right\| \le r_0 \\ \mu & \text{if} \quad \left\|\bar{\mathbf{X}} - \mu\right\| > r_0, \end{cases} \tag{159}$$

*in which*

$$r_0 = \max\left\{2M_p^{\frac{1}{p}}\sqrt{\frac{1}{m}\ln\frac{3(d+1)}{\nu}}, 4M_p^{\frac{1}{p}}(3m)^{\frac{1}{p}-1}\nu^{-\frac{1}{p}}\ln\frac{3(d+1)}{\nu}\right\}, \tag{160}$$

*then*

$$\left\|\mathbb{E}[\bar{\mathbf{X}}^*] - \mu\right\| \le C_p r_0 \nu \tag{161}$$

*for some constant $C_p$ that depends only on $p$.*

*Proof.* From (159),

$$\mathbb{E}[\bar{\mathbf{X}}^*] = \mathbb{E}[\bar{\mathbf{X}}\mathbf{1}(\left\|\bar{\mathbf{X}} - \mu\right\| \le r_0)] + \mu\mathrm{P}(\left\|\bar{\mathbf{X}} - \mu\right\| > r_0). \tag{162}$$

Thus

$$\mathbb{E}[\bar{\mathbf{X}}^*] - \mu = \mathbb{E}[(\bar{\mathbf{X}} - \mu)\mathbf{1}(\left\|\bar{\mathbf{X}} - \mu\right\| \le r_0)]. \tag{163}$$

Note that $\mathbb{E}[\bar{\mathbf{X}}] = \mathbb{E}[\mathbf{X}] = \mu$. Hence

$$\mathbb{E}[\bar{\mathbf{X}}^*] - \mu = -\mathbb{E}[(\bar{\mathbf{X}} - \mu)\mathbf{1}(\left\|\bar{\mathbf{X}} - \mu\right\| > r_0)]. \tag{164}$$

Therefore

$$\begin{aligned}
\left\|\mathbb{E}[\bar{\mathbf{X}}^*] - \mu\right\| &\le \mathbb{E}[\left\|\bar{\mathbf{X}} - \mu\right\|\mathbf{1}(\left\|\bar{\mathbf{X}} - \mu\right\| > r_0)] \\
&= \int_0^\infty \mathrm{P}(\left\|\bar{\mathbf{X}} - \mu\right\|\mathbf{1}(\left\|\bar{\mathbf{X}} - \mu\right\| > r_0) > t)dt \\
&= r_0\mathrm{P}(\left\|\bar{\mathbf{X}} - \mu\right\| > r_0) + \int_{r_0}^\infty \mathrm{P}(\left\|\bar{\mathbf{X}} - \mu\right\| > t)dt \\
&= r_0\nu + \sum_{k=0}^\infty \int_{r_k}^{r_{k+1}} \mathrm{P}\left(\left\|\bar{\mathbf{X}} - \mu\right\| > r_k\right)dt,
\end{aligned} \tag{165}$$

in which the last step uses Lemma 13, and

$$r_k = \max\left\{2M_p^{\frac{1}{p}}\sqrt{\frac{1}{m}\ln\frac{3(d+1)}{2^{-k}\nu}}, 4M_p^{\frac{1}{p}}(3m)^{\frac{1}{p}-1}(2^{-k}\nu)^{-\frac{1}{p}}\ln\frac{3(d+1)}{2^{-k}\nu}\right\}. \tag{166}$$

From Lemma 13, $\mathrm{P}(\left\|\bar{\mathbf{X}} - \mu\right\| > r_k) \le 2^{-k}\nu$, thus

$$\begin{aligned}
\left\|\mathbb{E}[\bar{\mathbf{X}}^*] - \mu\right\| &\le r_0\nu + \sum_{k=0}^\infty 2^{-k}\nu(r_{k+1} - r_k) \\
&= \nu\sum_{k=1}^\infty 2^{-k}r_k \\
&= r_0\nu\sum_{k=1}^\infty 2^{-k}\frac{r_k}{r_0}
\end{aligned} \tag{167}$$

From (166) and (160),

$$\frac{r_k}{r_0} \le 2^{\frac{k}{p}}\frac{\ln\frac{3(d+1)}{\nu} + k\ln 2}{\ln\frac{3(d+1)}{\nu}}, \tag{168}$$

thus

$$\begin{aligned}
\left\|\mathbb{E}[\bar{\mathbf{X}}^*] - \mu\right\| &\le r_0\nu\left[\sum_{k=1}^\infty 2^{--k(1-1/p)} + \frac{\ln 2}{\ln\frac{3(d+1)}{\nu}}\sum_{k=1}^\infty k2^{-k(1-1/p)}\right] \\
&\le C_p r_0\nu,
\end{aligned} \tag{169}$$

for some constant $C_p$ that depends only on $p$. The proof is complete. $\qquad\square$

