# OpenReview forum: "A Huber Loss Minimization Approach to Mean Estimation under User-level Differential Privacy"
_NeurIPS.cc/2024/Conference — NeurIPS 2024 poster_

### Official Review · Reviewer_zYzW · 2024-07-03

**Soundness:** 3
**Presentation:** 3
**Contribution:** 2
**Rating:** 5
**Confidence:** 3

**Summary:**

The paper proposes a user-level differentially private mechanism utilizing Huber loss minimization for mean estimation. This approach is robust to heavy-tailed distributions and addresses data imbalance across different users.

**Strengths:**

- The paper is well-written and easy to follow overall.
- The differentially private version of Huber loss minimization for mean estimation is an interesting contribution.

**Weaknesses:**

- It is unclear whether the convergence results (Theorem 5) are generally applicable to generic $w_i $'s. Using $m_i \land m_c$ seems somewhat unclear. What if the server (in federated learning) wants to compute the mean with $w_i = \frac{m_i}{\sum m_j}$?
- It would be beneficial to provide a detailed analytical comparison with WME, extending beyond Section G.
- An intuitive explanation of $\gamma$ is required. Does $\gamma$ measure imbalance? If so, how?
- The inequality in (10) appears to have a typo.
- $\lambda$ in Theorem 4 needs a definition.
- The figures are too small, making it difficult to check the results. What does the gray line in Figure 2(b) represent?

**Questions:**

See Weaknesses.

**Limitations:**

Limitation section is adequately provided

---

> ### Author Rebuttal · Authors · 2024-07-31
>
> Thanks for your valuable comments.
>
> # Reply to weaknesses
>
> **1. It is unclear whether the convergence results (Theorem 5) are generally applicable to generic $w_i$'s. Using $m_i\wedge m_c$ seems somewhat unclear. What if the server (in federated learning) wants to compute the mean with $w_i=m_i/\sum m_j$?**
>
> If we use $w_i=m_i/\sum m_j$, then the sensitivity will be too large if there are some users with large number of items. As a result, we must add a strong noise for privacy protection, which will hurt the performance. Using $m_i\wedge m_c$ ensures that the method is not severely affected by a single user with many items. This has been explained in line 299-305 in the paper.
>
> **2. It would be beneficial to provide a detailed analytical comparison with WME, extending beyond Section G.**
>
> Thanks for this comment. In Appendix G we have already shown that WME has worse rate than our method. The analysis in WME in [1] is based on $(\tau, \gamma)$-concentration. For our two main improvements, i.e. heavy-tailed distributions and imbalanced users, both the first and second one inevitably lead to large $\tau$.
>
> Here we list the our results and compare with WME here. For convenience, we omit the logarithm factors and non-private terms here.
>
> (1) Heavy-tailed, balanced users. For $n$ users with $m$ samples per user, under $p$-th order bounded moment,
>
> Ours: $\frac{d}{mn^2\epsilon^2}+(\frac{d}{m^2n^2\epsilon^2})^{1-1/p}$ (from eq.(14))
>
> WME: at least $\frac{d}{n^2\epsilon^2}(\frac{1}{m}+m^{4/p-2}n^{6/p})$ (from eq.(102))
>
> (2) Bounded support, imbalanced users. For $N$ total number of samples belonging to $n$ users,
>
> Ours: $\frac{d\gamma}{Nn\epsilon^2}$ (from eq.(20))
>
> WME: $\frac{d\gamma_0^2}{Nn\epsilon^2}$ (from eq.(104) multiplies a factor $d$)
>
> From the definition of $\gamma$ (Assumption 3) and $\gamma_0=nm_{max}/N$ (line 689), it is easy to prove that $\gamma_0>\gamma$. Moreover, the quadratic dependence $\gamma_0^2$ is reduced to linear dependence $\gamma$.
>
> We will make these comparison clearer in our revised paper.
>
> **3. An intuitive explanation of $\gamma$ is required. Does $\gamma$ measure imbalance? If so, how?**
>
> Yes, $\gamma$ measures the imbalance of users. Note that users are arranged in ascending order of $m_i$ (line 264-265). Therefore, from Assumption 3,  for users whose number of items are more than $\gamma$ times of the average number of items, the sum of items of these users are less than $1/2$. With small $\gamma$, users are nearly balanced, as the sizes of most of users are not much larger than the average sizes. On the contrary, with large $\gamma$, users are highly imbalanced. It means that the size of many users are much larger than the average size.
>
> Examples:
>
> (1) If users are balanced, then $\gamma=1$;
>
> (2) If the $i$-th user has $ki$ items (which means that the number of items of user is linear in its order), then for large $n$, $\gamma$ is approximately $\sqrt{2}$.
>
> **4. The inequality in (10) appears to have a typo.**
>
> Thanks for finding this typo. This should be a equal sign here.
>
> **5. $\lambda$ in Theorem 4 needs a definition.**
>
> Thanks. Here $\lambda$ is the smooth sensitivity $S(D)$. We will change the notation in the revised paper.
>
> **6. The figures are too small, making it difficult to check the results. What does the gray line in Figure 2(b) represent?**
>
> Thanks for this suggestion. We will make figures larger. In figure 2(b), we make a mistake of including other number of users without changing the legends. The corrected legends are:
>
> orange dashed curve -> WME, n=2000;
>
> brown dashed curve->WME, n=5000;
>
> gray dashed curve->WME, n=10000;
>
> red dashed curve-> WME, n=30000;
>
> blue solid curve-> HLM, n=2000;
>
> the two curves that appears to overlap at the bottom are HLM n=5000 and HLM n=10000, respectively.
>
> We refer to the global response for the corrected figure.
>
> # References
>
> [1] Levy et al. Learning with user-level privacy. NeurIPS 2021.

---

> > ### Comment · Reviewer_zYzW · 2024-08-10
> >
> > Thank you for the author's response. I believe the paper makes a reasonable contribution, and I will maintain my positive score.

---

### Official Review · Reviewer_BDM5 · 2024-07-08

**Soundness:** 2
**Presentation:** 2
**Contribution:** 3
**Rating:** 6
**Confidence:** 3

**Summary:**

The paper proposes a user-level differential private mean estimation based on minimizing weighted huber loss. The authors conduct theoretical and empirical assessments, showing that the proposed method is more robust to user-wise sample imbalance as well as heavy-tail distributions compared to the Winsorized mean estimator proposed in [0]

**Strengths:**

* The construction of smooth sensitivity and its analysis in both balanced and imbalanced user settings seem novel.
* The proposed method is thoroughly analyzed, providing both error upper bounds and empirical evaluations.

**Weaknesses:**

* Some claims in this paper might need more elaboration:
    > Line 57-58: “To the best of our knowledge, our method is the first attempt to unify robustness and DP at user-level”

    The definition of robustness should be clarified in the paper. Is it referring to robustness against heavy-tailed data, arbitrary outliers, or specific types of attacks? Therefore, it would be beneficial for the authors to explicitly define both robustness and outliers.

    > Line 93-94: “To the best of our knowledge, Huber loss minimization has not been applied to DP”

    Please see section 3.2.1 in [2]. Additionally, as a side note, gradient clipping for generalized linear loss is well-known to be connected to Huber loss [3].

* As the author also pointed out (line 347), the proposed algorithm requires the sample size of each local user as input to determine the Huber loss parameter $T$, which is a strong assumption.

**Questions:**

* Could the author compare the results in the balanced user case with existing literature, such as [1] and [4]? For example, a result similar to Theorem 2 has been derived in Theorem 4.1 of [1].
* What is the purpose of tuning $T_i$ in Section 7.2 when Theorem 5 already provides an optimal choice for $T_i$? Does this parameter tuning require an additional privacy budget?

**Limitations:**

The authors address the limitation of this work in section 8.



[0] Levy, Daniel, et al. "Learning with user-level privacy." Advances in Neural Information Processing Systems 34 (2021): 12466-12479.

[1] Narayanan, Shyam, Vahab Mirrokni, and Hossein Esfandiari. "Tight and robust private mean estimation with few users." International Conference on Machine Learning. PMLR, 2022.

[2] Avella-Medina, Marco, Casey Bradshaw, and Po-Ling Loh. "Differentially private inference via noisy optimization." The Annals of Statistics 51.5 (2023): 2067-2092.

[3] Song, Shuang, et al. "Evading the curse of dimensionality in unconstrained private glms." International Conference on Artificial Intelligence and Statistics. PMLR, 2021.

[4] Liu, Daogao, and Hilal Asi. "User-level differentially private stochastic convex optimization: Efficient algorithms with optimal rates." International Conference on Artificial Intelligence and Statistics. PMLR, 2024.

---

> ### Author Rebuttal · Authors · 2024-07-31
>
> Thanks for these valuable comments.
>
> # Reply to weaknesses
>
> **The definition of robustness should be clarified in the paper. Is it referring to robustness against heavy-tailed data, arbitrary outliers, or specific types of attacks? Therefore, it would be beneficial for the authors to explicitly define both robustness and outliers.**
>
> The robustness refers to arbitrary model poisoning attacks. The robustness of Huber loss minimizer has been already widely analyzed in existing works (In particular, [5] analyzes the robustness to Byzantine attacks in federated learning, which is relatively simpler compared with this paper). Therefore, we do not repetitively discuss the robustness of Huber loss minimizer in this paper. Following your comments, we will clarify these in our revised version.
>
> **Line 93-94: “To the best of our knowledge, Huber loss minimization has not been applied to DP”**
>
> **Please see section 3.2.1 in [2]. Additionally, as a side note, gradient clipping for generalized linear loss is well-known to be connected to Huber loss [3].**
>
> Thanks for bringing this paper into our attention. There are indeed some works that uses Huber loss minimization in DP. [2] discuss the linear regression problem. Despite different from ours, we will change the statement in the paper.
>
> Gradient clipping can be viewed as minimizing Huber loss, as is shown in Section 5.1 in [3]. This "Huber loss minimization" and ours have different meanings. In [3], the Huber loss approximates the population risk of DP optimization. In our paper, we do not focus on the optimization problem. Instead, we work on mean estimation, and there are no loss functions to approximate here.
>
> **As the author also pointed out (line 347), the proposed algorithm requires the sample size of each local user as input to determine the Huber loss parameter, which is a strong assumption.**
>
> Our experience is that it is common to assume the knowledge of sample sizes in distributed learning. For example, in [6], eq.(2), the weight is determined  by the proportion of samples of each client.
>
> While we agree that it is worthwhile to extend our work to the case that $m_i$ are also private, current setting is already practical. In distributed scenarios (especially federated learning), sample sizes of each client are usually not sensitive (see [7] for a review). It is fine if our knowledge of the sample size of local user is not very accurate. Our analysis can be easily generalized to the case in which we know an upper bound and a lower bound of the local sample size, such that the ratio of the upper bound to the lower bound is not large than a certain constant. Then rate of convergence of the overall mean squared error remains the same.
>
> # Reply to questions
>
> **Q1. Could the author compare the results in the balanced user case with existing literature, such as [1] and [4]? For example, a result similar to Theorem 2 has been derived in Theorem 4.1 of [1].**
>
> **Compared with [1] and [4], we (1) improve the performance for heavy-tailed distributions and (2) generalize to imbalanced users.** As discussed in line 225-227, for balanced users with bounded distributions, existing methods are already nearly optimal, and polynomial improvement is impossible. In our paper, the goal of Theorem 2 is to show that our improvement on heavy-tailed distributions and imbalanced users is not at the cost of hurting the performance under the simplest case with bounded distributions and balanced users.
>
> We will add discussions of these two papers in our revised version.
>
> **Q2. What is the purpose of tuning $T_i$ in Section 7.2 when Theorem 5 already provides an optimal choice for $T_i$? Does this parameter tuning require an additional privacy budget?**
>
> (1) In Theorem 5, $T_i$ is selected to minimize the theoretical upper bound. To ensure that the analysis is mathematically rigorous, the upper bound of estimation error is larger than the truth. Therefore, the optimal $T_i$ is different from that derived in theories.
>
> (2) The parameter tuning does not require additional privacy budget since in each experiment, $T_i$ are hyperparameters that is fixed before knowing the value of each sample. They are not determined adaptively based on the data.
>
> We will add these discussions in the revised version.
>
> # References
>
> [1] Narayanan, Shyam, Vahab Mirrokni, and Hossein Esfandiari. "Tight and robust private mean estimation with few users." International Conference on Machine Learning. PMLR, 2022.
>
> [2] Avella-Medina, Marco, Casey Bradshaw, and Po-Ling Loh. "Differentially private inference via noisy optimization." The Annals of Statistics 51.5 (2023): 2067-2092.
>
> [3] Song, Shuang, et al. "Evading the curse of dimensionality in unconstrained private glms." International Conference on Artificial Intelligence and Statistics. PMLR, 2021.
>
> [4] Liu, Daogao, and Hilal Asi. "User-level differentially private stochastic convex optimization: Efficient algorithms with optimal rates." International Conference on Artificial Intelligence and Statistics. PMLR, 2024.
>
> [5] Zhao, Puning et al. A huber loss minimization approach to byzantine robust federated learning. AAAI 2024. (Ref.[62] in the paper)
>
> [6] Wei, Kang, et al. "Federated learning with differential privacy: Algorithms and performance analysis." IEEE transactions on information forensics and security 2020.
>
> [7] Fu, Jie, et al. "Differentially private federated learning: A systematic review." arXiv:2405.08299.

---

> > ### Comment · Reviewer_BDM5 · 2024-08-11
> > **Official comment by reviewer BDM5**
> >
> > Thank you for answering my questions, particularly the one about Theorem 2.
> >
> > I have a further question:
> >
> > > "The parameter tuning does not require additional privacy budget since in each experiment ... They are not determined adaptively based on the data"
> >
> > Privacy leakage is still possible if the evaluation is solely on the training set. (e.g. mentioned in section 2 of https://arxiv.org/pdf/2110.03620)

---

> > > ### Author Response · Authors · 2024-08-11
> > >
> > > Thanks for introducing this paper. We have read [1], which is improved over an existing analysis of parameter tuning in [2].
> > >
> > > [1] Papernot, Nicolas, and Thomas Steinke. "Hyperparameter Tuning with Renyi Differential Privacy." ICLR 2022
> > >
> > > [2] Liu, Jingcheng, and Kunal Talwar. "Private selection from private candidates." STOC 2019.
> > >
> > > In our experiments, the parameter tuning does not lead to additional privacy leakage, since this is an experiment with synthesized data. After we change the parameters, samples are generated again. In other words, parameters for each experiment are determined before generating these samples. The parameters are not determined adaptively based on the data. Therefore there is no additional privacy leakage.
> > >
> > > [1] and [2] analyze the case of using a fixed dataset. When we update the hyperparameters, we have to reuse the dataset. As a result, the parameters have to be determined adaptively based on the data. As a result, the privacy leakage is inevitable.
> > >
> > > We will mention these discussions to avoid confusion.

---

> > > > ### Comment · Reviewer_BDM5 · 2024-08-13
> > > > **Official comment by reviewer BDM5**
> > > >
> > > > Thank you. Since my concerns have been resolved and considering the overall quality of the paper, I would like to increase my score to 6.

---

> > > > > ### Author Response · Authors · 2024-08-13
> > > > >
> > > > > Thank you very much for your response as well as the score increase! Please let us know if you have further questions or comments.

---

### Official Review · Reviewer_Bmi5 · 2024-07-08

**Soundness:** 3
**Presentation:** 3
**Contribution:** 3
**Rating:** 7
**Confidence:** 4

**Summary:**

Overall, the method proposed by the authors is interesting and effectively addresses the issue of privacy protection for users with imbalanced data. Compared to existing methods, the authors' approach is more robust and is supported by mathematical proofs.

**Strengths:**

1. The proposed method demonstrates significant innovation and holds substantial practical value.

2. The writing is clear, the arguments are well-structured, and the mathematical foundations are solid.

**Weaknesses:**

I have a generally positive view of this paper, though I have some questions and concerns that I hope the authors can address thoroughly.

**Questions:**

1. The authors' discussion lacks comprehensiveness. For instance, the statement "The most effective approach is the two-stage scheme, which finds a small interval first and then gets a refined estimate by clipping samples into the interval" is not fully substantiated. It would be beneficial to explore whether there are any end-to-end methods or approaches that integrate both stages. If such methods exist, a comparison should be provided. If they do not, an explanation should be offered as to why this research direction has not been pursued by others.

2. The authors should clearly explain why they chose the Huber loss, highlighting its advantages. Furthermore, they should clarify whether their proposed method involves any deep innovation beyond the simple use of the Huber loss, or if it merely employs the Huber loss without additional enhancements.

3. The authors' work bears similarities to "Private Mean Estimation with Person-Level Differential Privacy." The authors should discuss the distinctions between their work and this paper.

4. The authors should clarify whether their method has practical applications and specify the scale of data it can handle. Additionally, a complexity analysis of the method should be conducted.

---

> ### Author Rebuttal · Authors · 2024-07-31
>
> Thanks for your valuable comments.
>
> **Q1.The authors' discussion lacks comprehensiveness. For instance, the statement "The most effective approach is the two-stage scheme, which finds a small interval first and then gets a refined estimate by clipping samples into the interval" is not fully substantiated. It would be beneficial to explore whether there are any end-to-end methods or approaches that integrate both stages. If such methods exist, a comparison should be provided. If they do not, an explanation should be offered as to why this research direction has not been pursued by others.**
>
> The two-stage approach [1] is currently the most standard approach for user-level DP. To the best of our knowledge, before this work, there are no methods that integrate both stages. Follow-up research focus on different assumptions or different statistical problems, but they still use the localization-refinement two stage framework.
>
> Regarding why an end-to-end method is not pursued by others, we think that it is challenging to design a method that is adaptive to local sensitivity of data. It is necessary to finish a thorough analysis of the sensitivity that consider all possible cases. The user-wise mean concentrate around the true mean $\mu$ with high probability since the averaging operation within each user already reduces the variance. As a result, the local sensitivity is not large with high probability. However, extreme cases may happen, such that user-wise averages are far away from each other. Despite that these cases happen with low probability, the analysis of local sensitivity becomes significantly harder. Therefore, it is not straightforward to design an estimator that rigorously ensure $(\epsilon, \delta)$-DP. It has been mentioned in the paragraph 3 in the introduction. We will explain further in revision.
>
> **2.The authors should clearly explain why they chose the Huber loss, highlighting its advantages. Furthermore, they should clarify whether their proposed method involves any deep innovation beyond the simple use of the Huber loss, or if it merely employs the Huber loss without additional enhancements.**
>
> (1) Why Huber loss: Huber loss combines $\ell_2$ and $\ell_1$ loss, and strikes a tradeoff between bias and sensitivity. Huber loss minimizer is a widely used method for robust statistics. Moreover, robustness can be converted to DP.
>
> (2) Advantages: as has been discussed in multiple places in the paper, there are two advantages of our work: improved performance for heavy tailed distributions, and suitability to imbalanced datasets.
>
> (3) Innovation beyond the use of Huber loss. Huber loss was defined for scalars. In eq.(4) in the paper, Huber loss is defined for multi-dimensional vectors. Apart from the generalization to high dimensions, a more important technical novelty is the analysis of local sensitivity and the design of noise, which is the main challenge.
>
> **3.The authors' work bears similarities to "Private Mean Estimation with Person-Level Differential Privacy." The authors should discuss the distinctions between their work and this paper.**
>
> The paper [2] is indeed an important independent work that worths discussion. However, this paper is posted on arXiv after the NeurIPS submission deadline. Therefore, we are not aware of this paper at the time of submission.
>
> Actually [2] has already provided fruitful discussion in its updated version (see Section 1.3.1, arXiv 2405.20405). It has mentioned that [2] uses a directional bound, while we use a non-directional bound. Our assumption is slightly weaker than [2]. By some necessary rescaling, our Theorem 3 matches Theorem 4.1 in [2].
>
> We would like to comment further on the difference of methods. [2] still uses the two-stage method, with some refinements to handle the tails. Instead, our method is a direct Huber loss minimization approach, which is relatively easier to implement and requires less computation time. ([2] has not analyzed the time complexity. However, based on our understanding, the time complexity is not linear.)
>
> **4.The authors should clarify whether their method has practical applications and specify the scale of data it can handle. Additionally, a complexity analysis of the method should be conducted.**
>
> (1) Practical applications. As discussed in the last paragraph of the conclusion, for practical applications in federated learning, the remaining issue is that the method requires $n\gtrsim d$. $d$ is the number of model parameters here. In modern deep learning applications, $d$ is usually large, thus this condition is not satisfied. There are several potential solution: sparse DP mean estimation methods, and top-k gradient selection in federated learning.
>
> (2) Complexity. It has been discussed in line 154-158 in the paper. Further discussions have been provided in Appendix A.
>
>
> References
>
> [1] Levy et al. Learning with user-level privacy. NeurIPS 2021. (Ref.[27] in the paper)
>
> [2] Agarwal et al. Private Mean Estimation with Person-Level Differential Privacy. arXiv:2405.20405

---

> ### Comment · Reviewer_Bmi5 · 2024-08-08
>
> Thank you very much for the author's response, which has largely addressed my concerns. Overall, this paper demonstrates innovation and theoretical depth. Therefore, I will maintain a positive score. I hope the authors will include the discussed content in the final revised version to further improve the paper.

---

> > ### Author Response · Authors · 2024-08-09
> >
> > Thank you very much for your reply. If you have any remaining questions or suggestions for us, please let us know.

---

### Author Rebuttal · Authors · 2024-08-05

We thank all reviewers for the reviews. We are encouraged that reviewers have positive views on the mathematical solidness, practical value (Reviewer Bmi5), novelty (Reviewer BDM5) and presentation (Reviewer zYzW) of this paper.

The detailed feedbacks of each review are provided below. We are looking forward to your replies, so that we can engage in further discussions.

Regarding the weakness 6 raised by reviewer zYzW: There are some issues in Figure 2(b) in the paper. We have attached the revised figure here.

---

### Decision · Program_Chairs · 2024-09-25

**Decision:**

Accept (poster)

**Comment:**

While the reviewers are positive about the paper,  I (the AC) took a more close look at the paper. Huber loss minimization (and its connection to clipping) is not new in the DP literature. See Figure 1 in https://arxiv.org/pdf/2006.06783 . Also, see in the context of high dimensional linear regression (see https://proceedings.mlr.press/v30/Guha13.pdf#page=23.21). While the connection to federated learning with user level DP may be somewhat new, the paper is missing comparison to a reasonably large body of prior research. (See the citations in the papers mentioned.) In light of this, while we recommend acceptance, we also request the authors to carefully compare and disambiguate the claims made from prior research.